# Asymmetric CRISPR enabling cascade signal amplification for nucleic acid detection by competitive crRNA

Jeong Moon [1] & Changchun Liu [1] ✉

Nucleic acid detection powered by CRISPR technology provides a rapid, sensitive, and deployable approach to molecular diagnostics. While exciting, there remain challenges limiting its practical applications, such as the need for pre-amplification and the lack of quantitative ability. Here, we develop an asymmetric CRISPR assay for cascade signal amplification detection of nucleic acids by leveraging the asymmetric *trans*-cleavage behavior of competitive crRNA. We discover that the competitive reaction between a full-sized crRNA and split crRNA for CRISPR-Cas12a can induce cascade signal amplification, significantly improving the target detection signal. In addition, we find that CRISPR-Cas12a can recognize fragmented RNA/DNA targets, enabling direct RNA detection by Cas12a. Based on these findings, we apply our asymmetric CRISPR assay to quantitatively detect microRNA without the need for pre-amplification, achieving a detection sensitivity of 856 aM. Moreover, using this method, we analyze and quantify miR-19a biomarker in plasma samples from bladder cancer patients. This asymmetric CRISPR assay has the potential to be widely applied for simple and sensitive nucleic acid detection in various diagnostic settings.

Simple, sensitive, and accurate nucleic acid detection plays a critical role in early cancer diagnostics and infectious disease detection[1,2]. While PCR-based nucleic acid detection methods have been regarded as the gold-standard approach, due to their high sensitivity and specificity, there is a growing interest in developing alternative detection methods to address challenges such as the need for rapid, cost-effective, and easy-to-use diagnostic tools[1]. One such alternative is the CRISPR (clustered regularly interspaced short palindromic repeats)-Cas (CRISPR-associated proteins) system. Originally evolved as a bacterial immune system, CRISPR-Cas has been repurposed as a powerful tool for nucleic acid detection[2,3]. Various CRISPR-based methods have been developed to date, typically involving a pre-amplification step of target nucleic acids, followed by detection using CRISPR-Cas enzymes such as Cas12a or Cas13a[4,5]. Although the CRISPR-Cas system offers advantages in terms of simplicity and specificity, most CRISPR-based detection tools rely on separate pre-amplification and lack quantitative detection ability, limiting their practical applications[6–10].

Recently, many efforts have aimed to simplify CRISPR-based nucleic acid detection and improve its sensitivity and specificity. New strategies employ multiple crRNAs or catalytic nucleic acid circuits to avoid additional manipulation, yet these approaches require long targets capable of binding multiple crRNAs and have a long detection time[11,12]. Several studies have shown that chemically modified crRNA can promote improved gene correction compared to unmodified crRNA[13–15]. For example, some researchers reported that RNA-DNA hybrid crRNA and extensions on the 5' or 3' end of crRNA can activate the catalytic efficiency of CRISPR, thus enhancing the sensitivity and specificity of CRISPR-based nucleic acid detection[16–18]. Another study reported that a split crRNA containing separated scaffold and spacer RNA can catalyze highly specific *cis*-cleavage in Cas12a and, however, did not explore its effect on *trans*-cleavage[19].

CRISPR-Cas12a is an RNA-guided DNA endonuclease that can be programmed to cleave target DNA and has been harnessed for gene editing since its discovery[20,21]. In addition, Cas12a can induce

[1]Department of Biomedical Engineering, University of Connecticut Health Center, Farmington, CT 06032, US. ✉e-mail: chaliu@uchc.edu

indiscriminate cleavage of non-target single-stranded DNA (ssDNA) after target-specific recognition and cleavage, which is referred to as *trans*-cleavage activity[21,22]. Because of this property, Cas12a has been widely applied for nucleic acid-based molecular diagnostics[2,7]. However, since current Cas12a-based detection methods are limited to detecting double-stranded DNA (dsDNA) or ssDNA, a reverse transcription step and/or additional DNA pre-amplification steps are required for RNA detection[23–25]. In addition, CRISPR-Cas12a is typically activated by dsDNA target with a protospacer adjacent motif (PAM).

Here, we investigated asymmetric *trans*-cleavage behavior of Cas12a induced by the competitive reaction between full-sized crRNA and split crRNA and developed a cascade signal amplification method for nucleic acid detection, termed asymmetric CRISPR assay. We observed an asymmetric *trans*-cleavage reaction as well as a conformational resetting of the CRISPR enzyme resulting from competition between two types of crRNAs. We also demonstrated that this competitive CRISPR reaction could significantly enhance the detection sensitivity of CRISPR-Cas12a without the need for an additional DNA amplification step. Moreover, we found that Cas12a can recognize fragmented RNA/DNA targets, enabling direct detection of RNA. Based on these findings, we applied our asymmetric CRISPR assay to quantitatively detect a microRNA (miRNA) biomarker for liquid biopsy, demonstrating an amplification-free detection approach using CRISPR-Cas12a. Notably, this asymmetric CRISPR assay uses a single CRISPR-Cas12a enzyme and can achieve highly sensitive detection of miRNA, thereby showing potential as a powerful tool for simple, rapid, and highly sensitive nucleic acid-based molecular diagnostics.

## Results

### Asymmetric CRISPR assay

Figure 1 illustrates the working principle of the asymmetric CRISPR assay for cascade signal amplification detection of nucleic acid targets. Unlike the conventional CRISPR-Cas12a assay, which uses a single full-sized crRNA (Fig. 1a), our asymmetric CRISPR assay uses two sets of competitive crRNAs: i) a target-specific full-sized crRNA and ii) an independent split crRNA (Fig. 1b). The full-sized crRNA is designed to specifically bind to target nucleic acids (full-T). The split crRNA consists of a separate 5′-scaffold moiety (hereafter referred to as 'handle') and a 3′-spacer moiety (hereafter referred to as 'spacer'). The split crRNA is designed to specifically bind its own independent ssDNA (split-T) with a different sequence from the target nucleic acid. In the asymmetric CRISPR assay, when the two crRNAs (full-sized crRNA and split crRNA) and the split-T are mixed in one-pot, the full-sized crRNA will first form the Cas12a/full-sized crRNA complex because it has a stronger binding affinity to Cas12a compared with the split crRNA. Thus, the high-affinity full-sized crRNA inhibits the binding of the split crRNA to Cas12a and CRISPR-Cas12a cannot be activated even in the presence of the split-T (Fig. 1b). In the presence of target nucleic acids, the Cas12a/full-sized crRNA complex is specifically activated and initiates the first *trans*-cleavage reaction. Subsequently, the split crRNA can replace the full-sized crRNA and reactivate Cas12a for the second *trans*-cleavage reaction, which leads to an additional fluorescence signal amplification (Fig. 1b). Thus, the asymmetric CRISPR assay provides a simple, rapid, highly sensitive nucleic acid detection in one-pot by a single CRISPR-Cas12a.

### Competitive CRISPR reaction of full-sized crRNA and split crRNA

To investigate the binding affinity of different crRNAs to CRISPR-Cas12a, we designed a target-specific full-sized crRNA and a target-specific split crRNA that specifically binds to the same region of a DNA target. In addition, we designed two competitors (full-sized crRNA and split crRNA) that bind to a different target sequence than the target-specific full-sized crRNA and split crRNA (Fig. 2a). We compared the *trans*-cleavage activity of the target-specific split crRNA and full-sized crRNA at different concentrations of competitor full-sized crRNA (10, 20, 40, and 80 nM). We observed that the

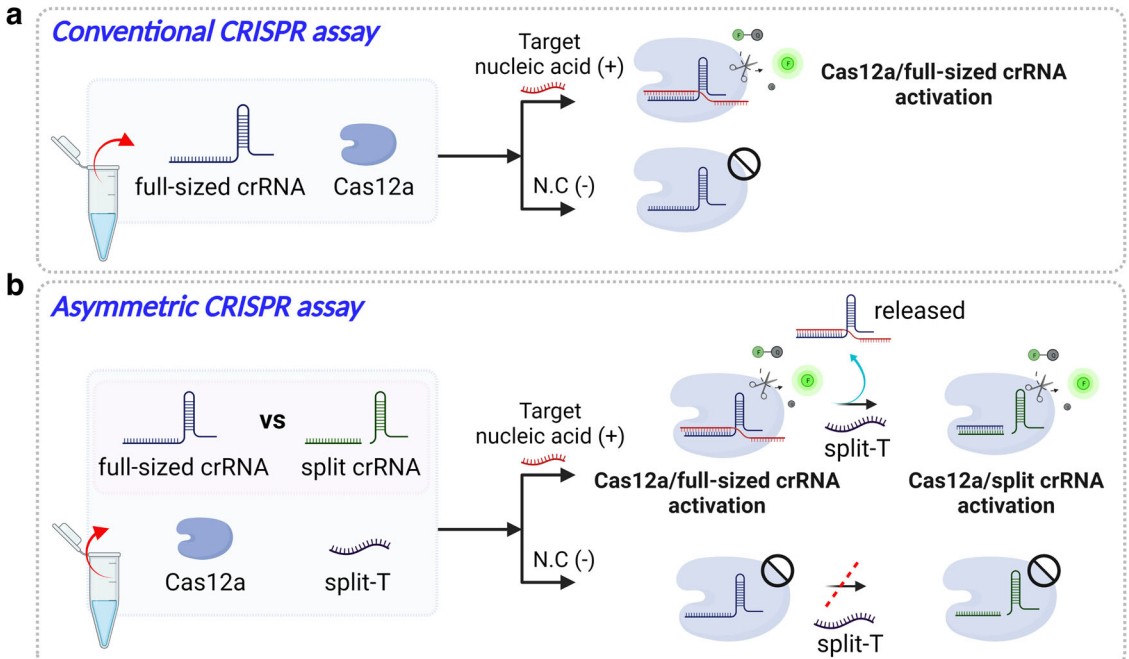

**Fig. 1 | Schematic illustration of nucleic acid detection by the conventional CRISPR assay and the asymmetric CRISPR assay. a** Schematic illustration of the conventional CRISPR-Cas12a assay by a single full-sized crRNA. **b** Working principle of the asymmetric CRISPR assay using two competitive crRNAs (full-sized crRNA and split crRNA). The full-sized crRNA has a higher affinity to Cas12a and specifically binds the target nucleic acid. The split crRNA is designed to bind to its own ssDNA sequence (split-T), which is different from the target nucleic acid. Due to the different binding affinities of the two crRNAs with CRISPR, CRISPR-Cas12a is first activated by the full-sized crRNA and its target nucleic acid and then is reactivated by the split crRNA and its split-T, resulting in cascade signal amplification of nucleic acid detection. N.C., negative control. Illustrations were created with BioRender.com.

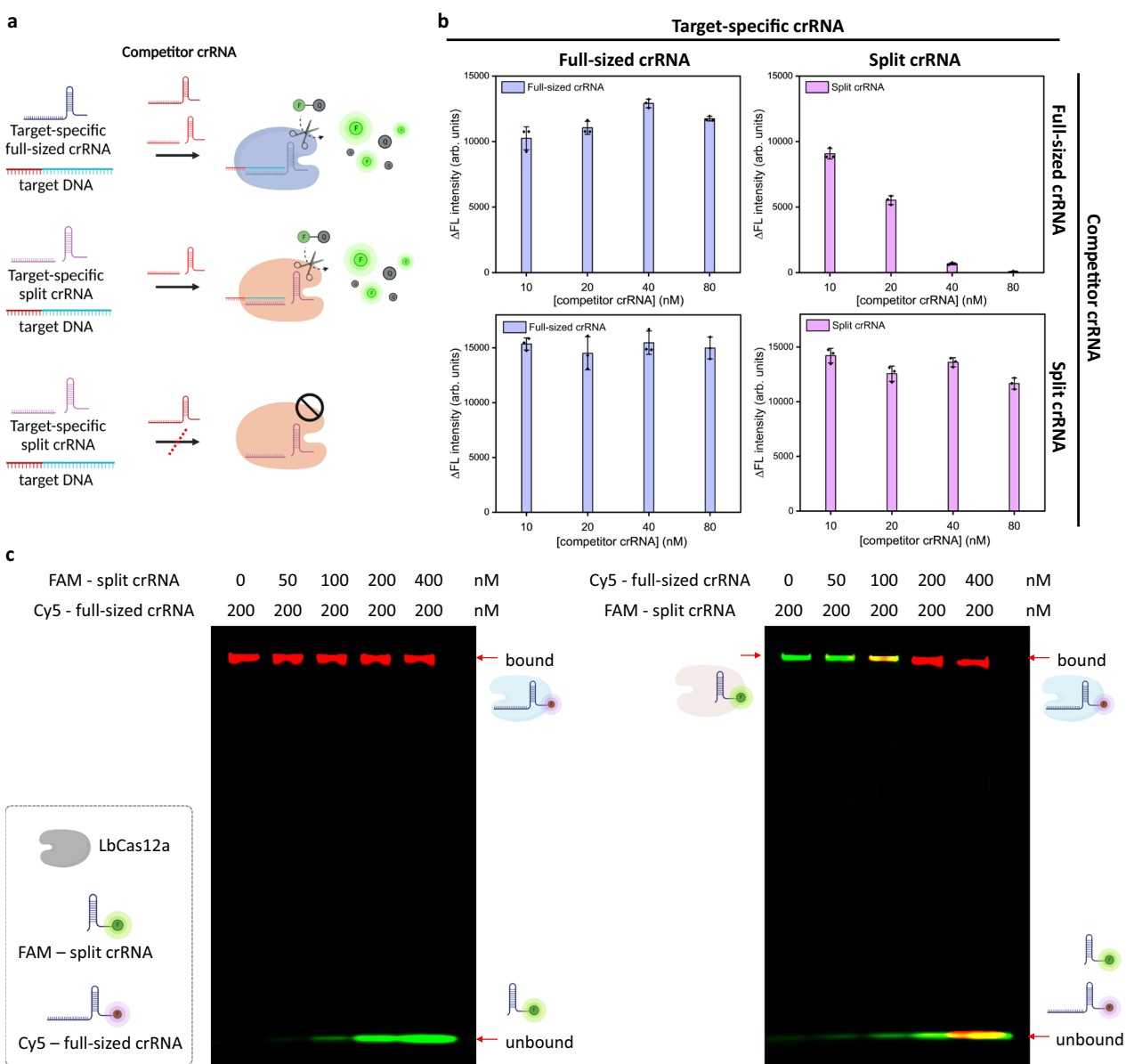

**Fig. 2 | Competitive CRISPR reaction between the full-sized crRNA and split crRNA. a** Effect of the competitor crRNA (full-sized and split crRNA) on the *trans*-cleavage reaction by target DNA and the target-specific crRNA (full-sized and split crRNA). The competitor crRNAs were designed to bind to different target sequences than the target-specific crRNAs. Illustration was created with BioRender.com. **b** ΔFluorescence intensity (ΔFL; $F_{Target\ DNA}$ - $F_{Control}$) of target-specific crRNA reactions in the presence of different concentrations of competitor crRNAs (10, 20, 40, and 80 nM). All the experiments were conducted in triplicate and graphs were represented by mean ± standard deviation (S.D). **c** Electrophoretic mobility shift assay. (Left) The concentration of Cy5-full-sized crRNA was fixed at 200 nM and the concentration of FAM-split-crRNA varied from 0 to 400 nM

(0, 50, 100, 200, and 400 nM). The Cy5-full-sized crRNA remained bound to Cas12a even when the split crRNA concentration was increased. (Right) The concentration of FAM-split-crRNA was fixed at 200 nM and the concentration of Cy5-full-sized crRNA varied from 0 to 400 nM (0, 50, 100, 200, and 400 nM). As the concentration of Cy5-full-sized crRNA increased, FAM-split crRNA bound to Cas12a was replaced with Cy5-full-sized crRNA. Each Gel image was measured by Cy5 and Alexa 488 fluorescence filters respectively, and then merged. The red band represents the Cy5 fluorescence signal, and the green band represents the FAM fluorescence signal. This experiment was performed twice. Source data are provided as a Source Data file.

*trans*-cleavage activity by the target-specific split crRNA significantly decreased as the concentration of the competitor full-sized crRNA increased. By contrast, the *trans*-cleavage reaction by the target-specific full-sized crRNA was not significantly affected as the concentration of the competitor full-sized crRNA increased (Fig. 2b). Next, we determined the effect of the competitor split crRNA on the *trans*-cleavage activity of the target-specific split crRNA and full-sized crRNA for DNA target detection. The competitor split crRNA did not significantly inhibit the *trans*-cleavage reactions of either the target-specific full-sized crRNA or the target-specific split crRNA (Fig. 2b). Based on these results, we

hypothesized that the full-sized crRNA has a stronger binding affinity to Cas12a than the split crRNA.

To test our hypothesis, we compared and visualized the interaction between Cas12a and crRNAs using an electrophoretic mobility shift assay. First, we designed Cy5-labeled full-sized crRNA and FAM-labeled split crRNA (handle). Next, we mixed the two types of crRNAs at different concentration ratios with Cas12a to form a Cas12/crRNA complex. As shown in Fig. 2c, the full-sized crRNA was highly bound to Cas12a even when the concentration of the split crRNA increased (left). On the contrary, the split crRNA lost its ability to bind to Cas12a as the concentration of the full-sized crRNA increased (right). These findings

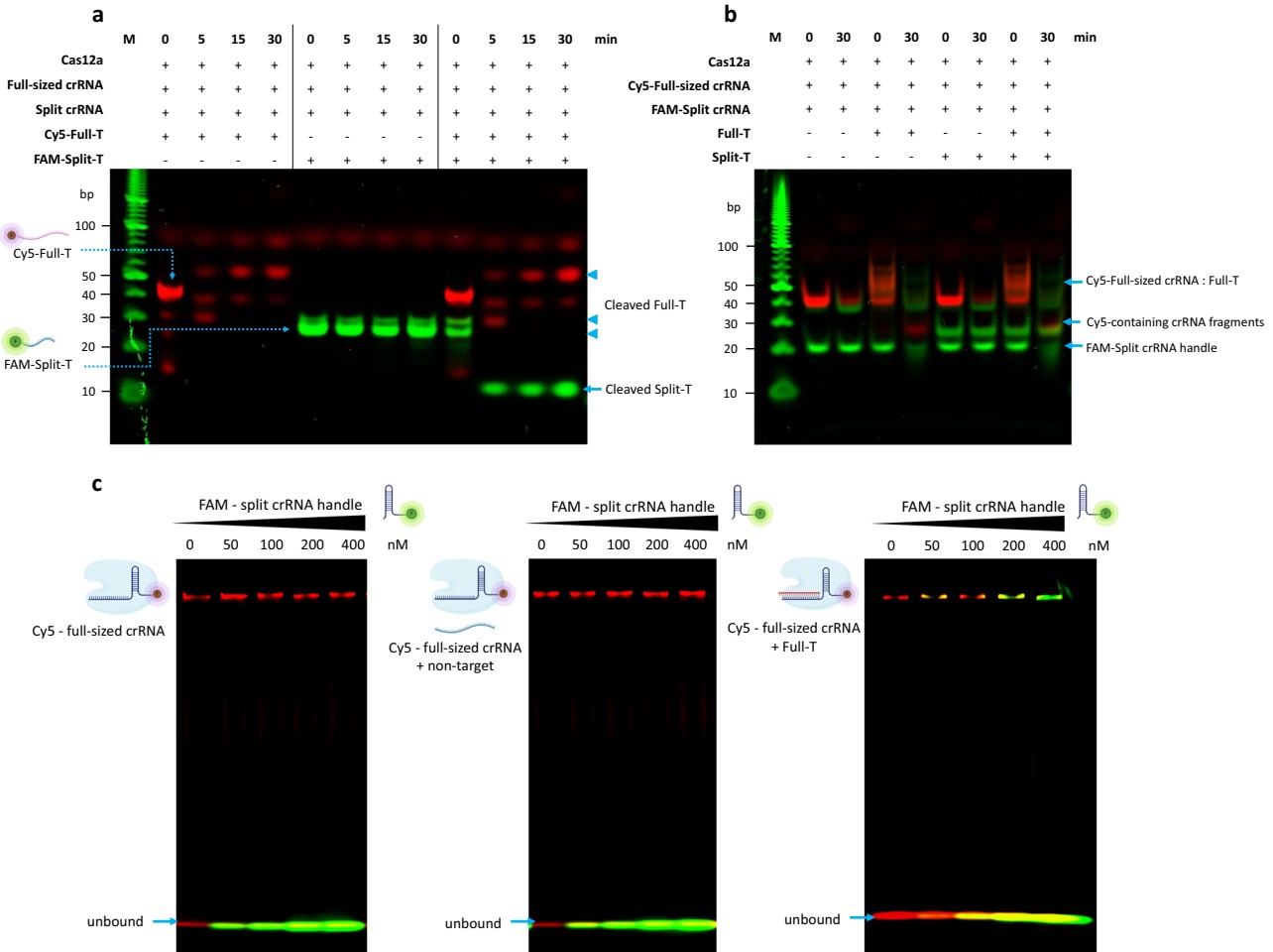

**Fig. 3 | Cascade signal amplification of nucleic acid detection by competitive crRNA. a** Cas12a, full-sized crRNA, and split crRNA were incubated with different combinations of Cy5-full-T and FAM-split-T for 0, 5, 15, and 30 min. More diverse control conditions were shown in Supplementary Figure 1. This experiment was performed twice. M = marker, [Cas12a] = 100 nM, [full-sized crRNA] = 40 nM, [split crRNA] = 10 nM, [Cy5-full-T] = 100 nM, and [FAM-split-T] = 50 nM. **b** Cas12a, Cy5-full-sized crRNA, and FAM-split crRNA were incubated with different combinations of full-T and split-T for 0 and 30 min. The 5′-Cy5-full-sized crRNA is trimmed at the 5′-end by LbCas12a (the first Cy5-conjugated-Uracil is cleaved).[17] More diverse control conditions were shown in Supplementary Figure 2. This experiment was performed twice. M = marker, [Cas12a] = 200 nM, [Cy5-full-sized crRNA, FAM-split

crRNA handle, and split crRNA spacer] = 100 nM, [full-T] = 100 nM, and [split-T] = 50 nM. **c** Electrophoretic mobility shift assay. The split crRNA binding to Cas12a is inhibited by the full-sized crRNA in the absence of the full-T (left) or in the presence of non-target (middle). The split crRNA can replace the full-sized crRNA-DNA hybrid or the cleaved R-loop in Cas12a after the full-sized crRNA/Cas12a is activated by the full-T (right). This experiment was performed twice. [LbCas12a] = 250 nM, [Cy5-full-sized crRNA] = 200 nM, [Full-T] = 100 nM, and [non-target] = 100 nM. Gel was measured by the Cy5 and Alexa488 filters, respectively, and then merged. The red band shows the Cy5 signal, and the green band shows the FAM/SYBR gold signal. Source data are provided as a Source Data file.

further support the fluorescence testing results caused by the *trans*-cleavage activity of CRISPR-Cas12a (Fig. 2b). Thus, the full-sized crRNA can regulate the split crRNA binding to Cas12a and generate a competitive CRISPR reaction due to its stronger binding affinity.

### Cascade signal amplification by competitive crRNA

By taking advantage of the difference in affinity to Cas12a between the full-sized crRNA and split crRNA, we next explored the cascade signal amplification mechanism for highly sensitive nucleic acid detection by conformational reactivation of CRISPR-Cas12a. In our asymmetric CRISPR assay, we used a larger amount of high-affinity full-sized crRNA than split crRNA. Thus, the affinity binding reaction with Cas12a by the full-sized crRNA proceeded dominantly and strongly while the reaction by the split crRNA was inhibited by the full-sized crRNA. We confirmed that the Cy5-full-T (target of full-sized crRNA) cleavage could occur by full-sized crRNA regardless of split crRNAs (Lanes 2 – 5, Fig. 3a); however, in the case of FAM-split-T (target of split crRNA), the cleavage reaction by split crRNA was inhibited in the presence of

full-sized crRNA (Lanes 6 – 9, Fig. 3a). Also, we further confirmed that FAM-split-T could be cleaved by split crRNA in the absence of the full-sized crRNA (Supplementary Figure 1a). These results indicate that the full-sized crRNA reaction predominantly occurs than the split crRNA reaction in the presence of both, which is consistent with the finding of the competitive CRISPR reaction between the two types of crRNAs (Fig. 2).

To better explain the mechanism of the competitive CRISPR reaction, we further used Cy5-full-sized crRNA and FAM-split crRNA to investigate the CRISPR reaction. In the presence of full-T, we observed that the hybrid structure of Cy5-full-sized crRNA and full-T gradually degraded, resulting in the generation of full-sized crRNA fragments containing Cy5 (Lanes 5 and 9, Fig. 3b). Also, we found that the 5′ end cleavage of crRNA even in the absence of target sequences (Fig. 3b and Supplementary Figure 2), which is consistent with the previous report[17]. Meanwhile, in the absence of full-T, the split crRNA reaction with Cas12a was initially suppressed by the full-sized crRNA despite the presence of split-T (Lanes 6 and 7, Fig. 3b). However, in the presence of

full-T, we observed a decrease in the signal of the FAM-split crRNA, which was thought to be due to the pre-crRNA processing by Cas12a (Lanes 5 and 9, Fig. 3b). In previous literature[26], a conformational resetting mechanism has been proposed to explain that new crRNA can replace the crRNA-DNA hybrid or the cleaved R-loop after the CRISPR-Cas12a cleavage reaction. Thus, based on our experimental results and previous literature[26], we hypothesized that the split crRNA could replace the full-sized crRNA and bind to Cas12a after the activation of full-sized crRNA, enabling a cascade signal amplification reaction of CRISPR.

To further test our hypothesis, we performed a competition assay in which pre-activated Cas12a/Cy5-full-sized crRNA, Cas12a/Cy5-full-sized crRNA/non-target, and Cas12a/Cy5-full-sized crRNA/full-T complexed were incubated with the increasing concentration of FAM-split crRNA. As shown in Fig. 3c, in the absence of the full-T or in the presence of non-target, the Cy5-full-sized crRNA remained bound to Cas12a and could not be displaced by split crRNA even with increasing concentration of split crRNA. Interestingly, FAM-split crRNA could bind to Cas12a under the condition of pre-activated Cas12a/Cy5-full-sized crRNA/full-T, which shows that the split crRNA can reset and activate Cas12a by replacing the hybrid of the full-sized crRNA and the full-T in Cas12a. In other words, when full-sized crRNA and split crRNA were mixed, the split crRNA reaction with Cas12a was initially suppressed by full-sized crRNA despite the presence of split-T. When the target nucleic acid of full-sized crRNA, full-T, was added, Cas12a was activated and triggered the first trans-cleavage reaction with full-sized crRNA. Then, the split crRNA could replace the full-sized crRNA and bind with Cas12a, reactivating Cas12a through its split-T and inducing a second trans-cleavage reaction, resulting in cascade signal amplification.

Next, we measured the fluorescence signal of target DNA at various concentrations by both the conventional CRISPR assay and the asymmetric CRISPR assay. As with the previous competitive crRNA reaction, we observed that the trans-cleavage reaction of the split crRNA for split-T decreased in the presence of the full-sized crRNA. Interestingly, when comparing the DNA target detection efficiency in the presence or the absence of the split crRNA and split-T, the detection efficiency was improved when the split crRNA and split-T were added (Supplementary Figure 3). While, when only either split crRNA handle or spacer was added, we did not observe the enhanced fluorescence signal (Supplementary Figure 4). In the case of DNA target detection, the asymmetric CRISPR assay detected the ssDNA target with a limit of detection (LOD) of 100 fM, which was 100 times more sensitive than that of the conventional CRISPR-Cas12a assay (Supplementary Figure 3). Thus, by leveraging the Cas12a kinetic difference caused by the competition reaction between two structurally different crRNAs, we were able to develop an asymmetric CRISPR assay for highly sensitive nucleic acid detection.

### RNA detection of Cas12a using fragmented RNA/DNA target

Cas12a/crRNA is known to catalyze trans-ssDNA cleavage after binding to a crRNA-complementary ssDNA as well as a dsDNA containing a PAM sequence[21]. For direct detection of RNA by CRISPR-Cas12a in our asymmetric CRISPR assay, we employed two fragmented nucleic acid targets that specifically bind crRNA and unleash indiscriminate trans-ssDNA cleavage activity of Cas12a (Fig. 4a). We investigated the trans-cleavage reaction after assembling Cas12a/crRNA with an ssDNA activator and ssRNA/ssDNA target in different combinational fashions (Fig. 4b–d). The trans-cleavage activity cannot occur when only the ssDNA activator or ssRNA/ssDNA target is present. Interestingly, the ssDNA 5'-activator/ssDNA 3'-target and ssRNA 5'-target/ssDNA 3'-activator were able to induce the collateral cleavage of Cas12a, whereas the ssDNA 5'-activator/ssRNA 3'-target could not. This result could be attributed to the weak binding affinity of the ssRNA 3'-target with crRNA compared with the ssDNA 3'-activator when it is targeted to the crRNA's seed region. Recently, several studies reported that a seed region that is 5-10 nucleotides (nt) distances from the PAM within the protospacers is important for target recognition of Cas12a[27–30]. In addition, a similar finding was reported that the PAM-proximal 'seed' region of crRNA strictly tolerates DNA for initiating trans-cleavage, and the PAM-distal region or 3'-end of the crRNA can tolerate both RNA and DNA substrates[31]. Thus, the binding position of ssRNA with crRNA highly affects the trans-cleavage activity of Cas12a (Fig. 4a–d).

After validating the RNA detection ability of Cas12a by using fragmented RNA/DNA targets, we next attempted to apply it to detect miRNA. As a proof of concept, we selected miR-19a as a target miRNA sequence because it is a potential biomarker for early diagnostics of many cancers. We designed a universal ssDNA 3'-activator complementary to the seed region of the crRNA. To compare and optimize miR-19a detection, we designed a series of crRNAs to bind ssDNA 3'-activator/miR-19a 5'-target with a single-nucleotide difference (Supplementary Figure 5). The crRNA that binds to miRNA 5'-target and ssDNA 3'-activator is hereafter referred to as RD crRNA. As shown in Supplementary Figure 5, the binding length between RD crRNA and ssDNA 3'-activator was highly related to the trans-cleavage activity of Cas12a. As the binding length between RD crRNA and ssDNA 3'-activator increased, the trans-cleavage reaction was strongly induced; however, it was reduced as the binding length decreased to 8-9 nt.

For comparison, we designed a miR-19a 3'-target/ssDNA 5'-activator by reversing the targeted locations of the RD crRNA and designed crRNA that binds to ssDNA 5'-activator and miR-19a 3'-target (DD crRNA). Interestingly, they could not initiate the collateral cleavage reaction of Cas12a regardless of the crRNA binding position (Supplementary Figure 6). By contrast, when both target nucleic acids were ssDNA, we observed an efficient cleavage reaction regardless of the binding length (Supplementary Figure 7). Based on this finding, we placed the miRNA at the 5'-target position close to the 3' end of the crRNA. In addition, we designed an RD crRNA to bind to 9 nt of the miRNA and to 12 nt of the DNA activator. Furthermore, we tested the detection sensitivity of Cas12a for miRNA detection by using fragmented RNA/DNA targets. The real-time fluorescence signal was measured depending on various concentrations of target miRNA (100 fM ~ 1 nM) and confirmed that a 1 pM level of miRNA could be detected (Supplementary Figure 8). When compared to the detection efficiency of a full-sized ssDNA target containing miRNA and DNA activator sequences, there was no significant difference between the ssDNA target and the miRNA/DNA hybrid targets (Supplementary Figure 9). This finding also indicates that the RNA target could initiate the trans-cleavage activity of Cas12a by using a fragmented RNA/DNA target strategy and achieved a similar degree as the conventional full-sized ssDNA target.[32]

### Amplification-free miRNA quantitative detection

Next, we investigated whether we could further improve the miRNA detection sensitivity of Cas12a by applying our asymmetric CRISPR assay, enabling amplification-free miRNA detection. To eliminate the need for an additional split crRNA target, we designed the ssDNA 3'-activator that simultaneously can serve as the split crRNA target. Here, we first designed a split crRNA to recognize part of the DNA activator (blue portion of the DNA activator, Fig. 5a), along with a full-sized crRNA that simultaneously binds to the miRNA (red portion of the miRNA, Fig. 5a) and the DNA activator (red portion of the DNA activator, Fig. 5a). To determine the optimal conditions for the asymmetric CRISPR assay, we first tested the concentration of full-sized crRNA and split crRNA (Fig. 5b, c). The fluorescence signal was measured from each target sample and control sample, and then the ΔFL intensity was obtained by subtracting the fluorescence signal of the control sample from the fluorescence signal of the target sample. Since the split crRNA reaction is inhibited by the full-sized crRNA, the fluorescence signal was not changed significantly with the change in split crRNA concentrations

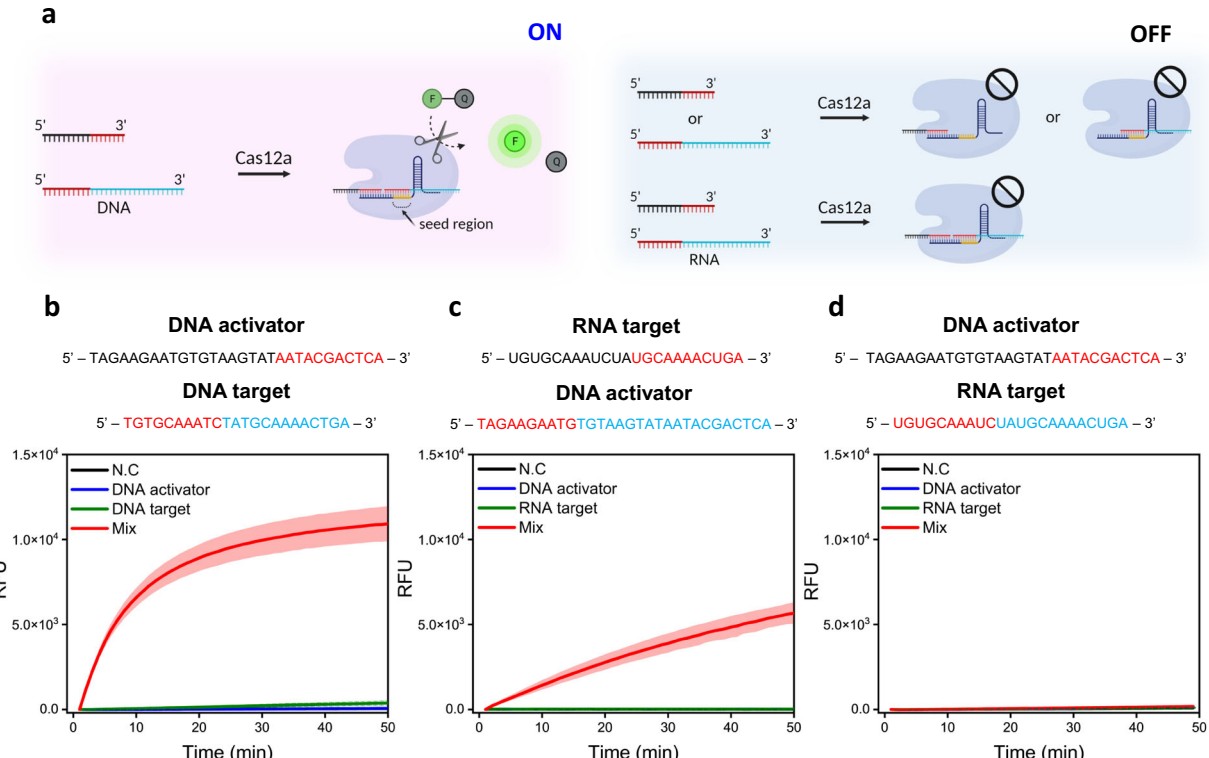

**Fig. 4 | Comparison and optimization of RNA detection of Cas12a using fragmented nucleic acid target. a** Schematic illustration of the Cas12a/crRNA complex binding two fragmented single-stranded nucleic acid targets (black-red and red-blue, respectively, with red indicating the crRNA binding region), resulting in activation of a *trans*-cleavage reaction. The *trans*-cleavage reaction cannot be induced if only one of the fragmented targets is present. Illustration was created with BioRender.com. **b**–**d** Sequences of the DNA activator and fragmented nucleic acid target (upper b-d, red indicates the crRNA binding region), and the corresponding fluorescence signal under different conditions (below b-d, N.C, activator, target, and the mixture of activator and target). Cas12a ribonucleoprotein complex made with DD crRNA was tested for b and d, and that made with RD crRNA was tested for c. All the experiments were conducted in triplicate and graphs were represented by mean (bold line) ± standard deviation (S.D). [Cas12a] = 100 nM, [DD and RD crRNA] = 10 nM, [DNA activator] = 100 nM, [DNA target] = 100 nM, [RNA target] = 100 nM. Source data are provided as a Source Data file.

ranging from 10 to 40 nM. In the case of full-sized crRNA, the most optimal results were observed at a concentration of 40 nM. As shown in Fig. 5b–e, the conditions for obtaining optimal detection performance were determined to be 100 nM for LbCas12a, 40 nM for full-sized crRNA, 10 nM for split crRNA, and 20 nM for DNA activator. When testing different targets, the reaction condition can be further optimized due to the potential existence of the secondary or tertiary structure of different target sequences. Under the optimized reaction conditions, we monitored real-time fluorescence signals for miRNA detection at various concentrations ranging from 1 fM to 1 nM. Interestingly, in the high-concentration target miRNA condition, there was no significant difference compared with the CRISPR detection without split crRNA; however, a notable difference in fluorescence signal emerged in the low-concentration target condition (Fig. 5f, g, and Supplementary Figure 10). For instance, the endpoint fluorescence signal difference of the 1 pM target miRNA in the asymmetric CRISPR assay was 29.2 times higher than that in the absence of split crRNA (Fig. 5f).

As shown in Fig. 5g, ΔFluorescence intensity ($F_{target \ miRNA}$ − $F_{control}$), which is the difference in fluorescence signal between the control and target miRNA, was improved by adding split crRNA, and the detection sensitivity was significantly improved compared to the case without split crRNA. In particular, the fluorescence intensity linearly increased with the logarithm concentration of miR-19a ranging from 1 fM to 10 pM in the presence of split crRNA. With our asymmetric CRISPR assay, we could achieve a LOD of 856 aM of miR-19a, which is 1,000 times more sensitive than the CRISPR detection in the absence of split crRNA (Supplementary Figure 10, Supplementary Table 1). This detection sensitivity is comparable to the CRISPR-based miRNA

detection methods combined with pre-amplification step[23–25]. To demonstrate the detection versatility of the assay, we also tested let-7a, another cancer miRNA biomarker, and obtained a similar result (Supplementary Figure 11). Next, to evaluate the selectivity of our developed assay, we tested and compared several types of miRNAs (let-7a, miR-21, miR155, and miR-122). We observed a strong fluorescence increase with the target miR-19a; however, other types of miRNAs were unable to enhance the fluorescence signal, indicating the high specificity of this assay (Fig. 5h and i). Thus, the asymmetric CRISPR assay provides a simple, highly sensitive, and specific approach for amplification-free, quantitative detection of miRNA.

**Clinical validation of the asymmetric CRISPR assay**
Finally, we explored the potential utility of the asymmetric CRISPR assay for miRNA liquid biopsy in cancer diagnosis and prognosis. miRNAs have been implicated in the development and progression of various types of cancer[33]. miRNA liquid biopsies that detect changes in miRNA expression levels in bodily fluids (e.g., blood, urine, or saliva) can provide important diagnostic and prognostic information about diseases and are more convenient and patient-friendly than tissue biopsies[34]. In particular, studies have shown that miR-19a is over-expressed in bladder cancer patients, contributing to the growth and metastasis of bladder cancer[35,36]. Here, we analyzed the expression level of miR-19a in plasma samples collected from patients with bladder cancer using the asymmetric CRISPR assay (Fig. 6a).

First, we extracted total RNA from plasma samples collected from ten bladder cancer patients and five healthy donors using a commercial RNA extraction kit. We mixed the extracted total RNA with a

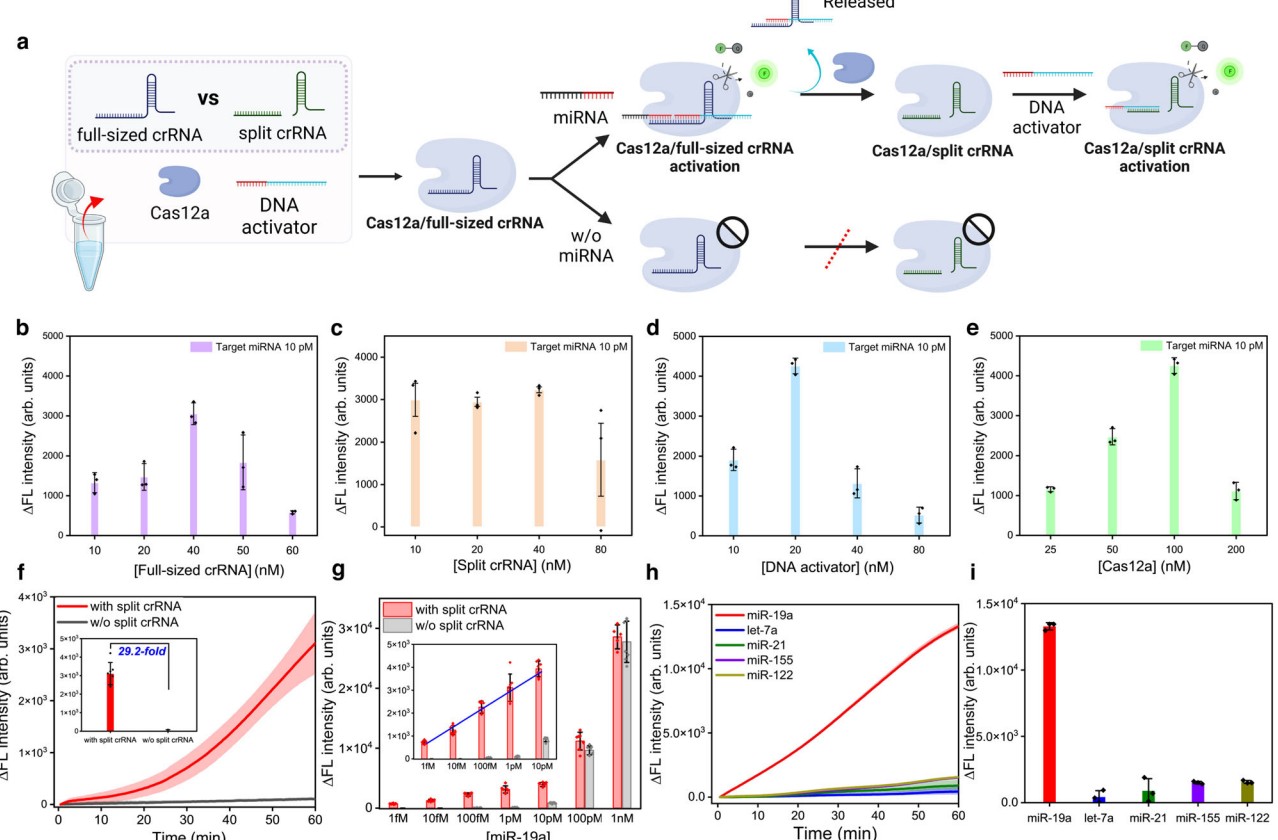

**Fig. 5 | miRNA quantitative detection by the asymmetric CRISPR assay.**
**a** Schematic illustration of miRNA detection based on the asymmetric CRISPR assay. Illustration was created with BioRender.com. **b** ΔFluorescence intensity (F$_{Target\ miRNA}$ − F$_{control}$) depending on the full-sized crRNA concentration from 10 to 60 nM. [Cas12a] = 100 nM, [split crRNA] = 10 nM, [DNA activator] = 20 nM. ($n$ = 3, Data are represented as mean ± S.D of three technical replicates). **c** ΔFluorescence intensity depending on the split crRNA concentration from 10 to 80 nM. [Cas12a] = 100 nM, [RD crRNA] = 40 nM, [DNA activator] = 20 nM. ($n$ = 3, Data are represented as mean ± S.D of three technical replicates). **d** ΔFluorescence intensity depending on the DNA activator concentration from 10 to 80 nM. [Cas12a] = 100 nM, [split crRNA] = 10 nM, [RD crRNA] = 40 nM. ($n$ = 3, Data are represented as mean ± S.D of three technical replicates). **e** ΔFluorescence intensity depending on the Cas12a concentration from 25 to 200 nM. [split crRNA] = 10 nM, [RD crRNA] = 40 nM, [DNA activator] = 20 nM. ($n$ = 3, Data are represented as mean ± S.D of three technical replicates) miR-19a sequence was used as a target, and the concentration

was 10 pM. **f** Time-dependent fluorescence signal changes of the asymmetric CRISPR assay (red) and Cas12a/crRNA reaction (black). Inset is the Δfluorescence intensity (F$_{Target\ miRNA}$ − F$_{control}$) comparison of the asymmetric CRISPR assay (red) and Cas12a/crRNA reaction (black). The target miR-19a concentration is 1 pM. Each curve was subtracted by the control signal. ($n$ = 7 and data represent mean ± S.D of seven technical replicates). **g** ΔFluorescence intensity of the asymmetric CRISPR assay (red) and Cas12a/crRNA (black) as a function of the target miRNA concentrations. Inset is the linear relationship between the fluorescence signal and logarithm concentration of miR-19a ranging from 1 fM to 10 pM. ($n$ = 7 and data represent mean ± S.D of seven technical replicates). **h, i** Time-dependent fluorescence signal and Δfluorescence intensity of different types of miRNAs (miR-19a, let-7a, miR-21, miR-155, and miR-122). [miR-19a] = 100 pM, [let-7a, miR-21, miR-155, and miR-122] = 1 nM. ($n$ = 3, Data are represented as mean ± S.D of three technical replicates) Source data are provided as a Source Data file.

reaction solution including Cas12a/crRNA and split crRNA, then measured the fluorescence signal in real-time. Subsequently, we estimated the concentration of the target miR-19a in each plasma sample based on the standard curve that we obtained using the relationship between miRNA concentration and the ΔFluorescence intensity (F$_{target\ miRNA}$ − F$_{control}$). As a result, we confirmed that the overall expression level of miR-19a was higher in the bladder patient samples compared to the healthy donor samples (Fig. 6b and d). Concurrently, we analyzed the same extracted RNA samples using reverse-transcription quantitative PCR (RT-qPCR). We first constructed a standard curve (Supplementary Figure 12) based on the relationship between miRNA concentration and the Cq values of the RT-qPCR. Then, we calculated the concentration of miR-19a in each plasma sample based on the standard curve of the RT-qPCR. Our developed assay showed an excellent correlation with the results using RT-qPCR as shown in Fig. 6c and d. Together, these clinical testing results demonstrate that the asymmetric CRISPR method enables a simple, rapid, and amplification-free detection of miRNA in clinical

samples with high sensitivity, providing empirical support for its clinical application.

## Discussion

In summary, we see two major CRISPR findings in this study: (i) the competitive reaction between full-sized crRNA and split crRNA can improve the detection sensitivity of CRISPR-Cas12a (Figs. 2 and 3), and (ii) Cas12a can recognize fragmented RNA/DNA targets that are complementary to the crRNA, enabling direct detection of RNA (Fig. 4). Based on these findings, we developed an asymmetric CRISPR assay for cascade signal amplification detection of nucleic acid and applied it to quantitatively detect miRNA in clinical cancer samples without the need for pre-amplification or reverse transcription.

A previous study reported that split crRNA can catalyze highly specific and efficient cleavage of target DNA by Cas12a nucleases in vitro and in lysates of human cells[19]. However, little is known about the *trans*-cleavage activity of split crRNA. In this study, we found an interesting *trans*-cleavage behavior between split crRNA and full-sized

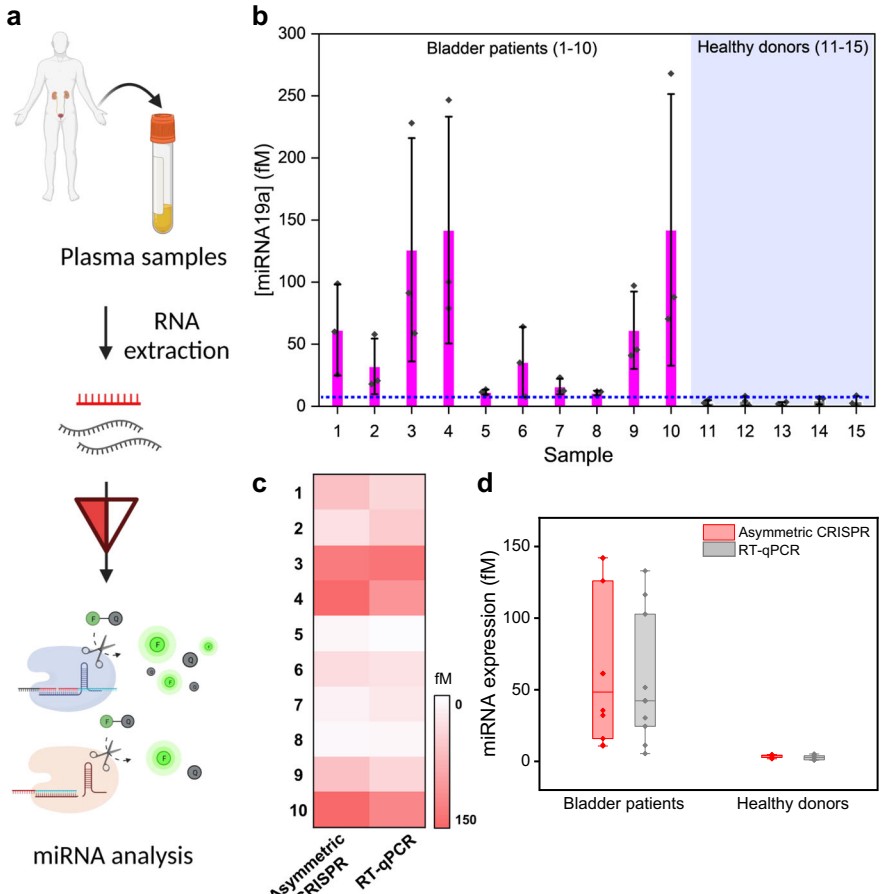

**Fig. 6 | Clinical application of the asymmetric CRISPR assay for amplification-free miRNA detection in clinical samples. a** Schematic illustration of miRNA detection in human plasma samples using the asymmetric CRISPR assay. Illustration was created with BioRender.com. **b** Estimated target miR-19a concentrations in the bladder patient plasma samples (samples 1–10) and the healthy donor plasma samples (samples 11–15). The Blue dashed line represents the highest target miRNA concentration of healthy donors plus standard deviation. ($n = 3$, Data are represented as mean ± S.D of three technical replicates.) **c** Heatmap of the estimated target miR-19a concentrations of the asymmetric CRISPR assay and conventional RT-qPCR assay. **d** miR-19a expression level analyzed by asymmetric CRISPR (red) and RT-qPCR (black) in bladder patients and healthy donors. The center line represents the median expression level, the bounds of the box indicate the interquartile range, and the whiskers indicate the maximum and minimum values. Source data are provided as a Source Data file.

crRNA. Our results indicate that full-sized crRNA has a stronger binding affinity to Cas12a than split crRNA, which enables full-sized crRNA to regulate the split crRNA reaction. Specifically, when two types of crRNAs with different affinities to Cas12a are mixed, the *trans*-cleavage reaction caused by the crRNA showing a stronger affinity for Cas12a is fast and dominant. By contrast, the reaction induced by the crRNA with a weaker affinity for Cas12a is slow but helps to enhance the sensitivity of the dominant reaction. Leveraging these specific asymmetric *trans*-cleavage behaviors of competitive crRNA, we developed a signal amplification method that can improve the sensitivity of CRISPR simply by adding split crRNA and its target ssDNA. To our knowledge, this is the first report of a signal amplification method using competitive crRNAs with different CRISPR reactivities.

The *class*-II type V CRISPR-*Cas12a* system is known to recognize dsDNA and ssDNA activators and exhibits collateral nonspecific ssDNA cleavages. In this study, we revealed that Cas12a can directly recognize RNA targets when the RNA is placed at the 3' end of the crRNA and supported by a DNA located at the 5' end of the crRNA. In particular, the binding region of the RNA to the crRNA is important, as the ssDNA must be located at the 5' end of the crRNA, a seed region that plays a critical role in target recognition and cleavage by the Cas12a protein. A recent report showed that Cas12a can be programmed to bind ssRNA, supporting our findings[31]. More importantly, these results indicate that Cas12a can be programmed to detect both RNA and DNA.

Furthermore, we adapted our asymmetric CRISPR assay to quantitatively detect miRNA biomarker by combining fragmented RNA/DNA targets. In our assay, we designed the split crRNA to bind a DNA activator that supports RNA detection; as a result, target miRNA was selectively detected with a LOD of 856 aM (1,000-fold more sensitive than the CRISPR detection without split crRNA), in line with previous studies combining Cas12a with pre-amplification steps[23–25]. Moreover, we demonstrated that our assay can specifically detect miRNA in human plasma samples, showing its potential application for liquid biopsy. Compared with the current CRISPR-Cas system-based miRNA detection methods (Supplementary Table 2), the developed method is a one-pot, one-step isothermal signal amplification method using a single CRISPR enzyme, enabling simple, sensitive, and quantitative detection of nucleic acids. In addition, the asymmetric CRISPR is cost-effective with the material cost estimated to be as low as US$0.47 per test (Supplementary Table 3).

Despite the promising results, the asymmetric CRISPR assay can be further expanded and improved from several aspects in future studies, enabling point-of-care testing in resource-limited environments[37,38]. First, to simplify the manual operation, the assay could be further integrated into a microfluidic platform for automated sample preparation and liquid manipulation[39]. Second, to improve single-nucleotide sequence discrimination of the assay, newly engineered CRISPR enzymes and engineered crRNAs can be exploited and

optimized[15,40]. Third, to eliminate the need for expensive fluorescence detection equipment, a simple and affordable smartphone detector can be developed for fluorescence recording and signal processing. Last, beyond the current miRNA detection for early cancer diagnostics, the asymmetric CRISPR assay can be further developed to detect other nucleic acid biomarkers of infectious diseases, such as SARS-CoV-2, and HIV[41–44]. Overall, this study highlights the discoveries of the CRISPR-Cas12a reaction system and presents a powerful nucleic acid detection approach with the potential for clinical applications for early cancer diagnostics and infectious disease detection.

# Methods

## Ethical statement

Human plasma samples from healthy donors were purchased from Innovative Research (USA). Human plasma samples from bladder cancer patients were collected and provided by the Carole and Ray Neag Comprehensive Cancer Center with a protocol approved by the ethics committee at the University of Connecticut Health Center (IRB #08-310-1).

## Materials

All oligonucleotide sequences used in this study were obtained from Integrated DNA Technologies (IDT, USA) and are listed in Supplementary Table 4. LbCas12 (Cpf1, Cat# M0653), 10× NEBuffer 2.1 (Cat# B6002S), and proteinase K (Cat# P8107S) were purchased from New England Biolabs (NEB, USA). RNase-free water (Cat# 11-05-01-14) was purchased from IDT. The 10× TBE (tris borate EDTA) buffer (Cat# 1610770), 10× TAE (tris acetate EDTA) buffer (Cat# 161-0743), 40% acrylamide/bis solution (Cat# 1610146), ammonium persulfate (Cat# 1610700), and TEMED (Cat# 1610800) were purchased from Bio-Rad (USA). SYBR gold nucleic acid gel stain was purchased from Invitrogen (USA). Human plasma samples from healthy donors were purchased from Innovative Research (USA). Human plasma samples from bladder cancer patients were collected and provided by the Carole and Ray Neag Comprehensive Cancer Center with a protocol approved by the ethics committee at the University of Connecticut Health Center (IRB #08-310-1) (Supplementary Table 5). The miRNeasy Serum/Plasma Advanced Kit (Cat# 217204), miRCURY LNA RT Kit (Cat# 339340), miRCURY LNA SYBR Green PCR Kit (Cat# 339345), and has-miR-19a-3p miRCURY LNA miRNA PCR assay (Cat# 339306) were purchased from Qiagen (USA).

## Competitive CRISPR reaction assay between the full-sized crRNA and split crRNA

2 µL LbCas12a (1 µM), 0.2 µL target-specific full-sized crRNA (1 µM) (0.2 µL each of 1 µM target-specific split crRNA handle and spacer), 0.2 µL of different concentrations of competitor full-sized crRNA (0.2 µL each of competitor split crRNA handle and spacer), 1 µL fluorescence-quencher probe (10 µM) and 2 µL 10X NEBuffer r2.1 buffer was mixed with 2 µL ssDNA target (10 nM) and added RNase-free water to a final volume of 20 µL. For the control sample, 2 µL RNase-free water was added instead of the ssDNA target. Each fluorescence signal of the target and control sample was measured in real-time at 37 °C using a CFX96 touch real-time PCR system (Bio-Rad, CA, USA).

## Conventional Cas12a/crRNA assay

The cleavage reaction was conducted in a final volume of 20 µL, including 2 µL 10X NEBuffer r2.1 buffer, 2 µL LbCas12a (1 µM), 0.2 µL crRNA (1 µM), 1 µL fluorescence-quencher probe (10 µM), 12.8 µL RNase-free water and 2 µL various concentrations of target ssDNA. The fluorescence signal was measured at 30-s intervals at 37 °C using a CFX96 touch real-time PCR system (Bio-Rad, CA, USA).

## Fragmented nucleic acid target detection using Cas12a/crRNA

The cleavage reaction was conducted in a final volume of 20 µL, including 2 µL 10X NEBuffer r2.1 buffer, 2 µL LbCas12a (1 µM), 0.2 µL crRNA (1 µM), 2 µL DNA activator (100 nM), 1 µL fluorescence-quencher probe (10 µM), 10.8 µL RNase-free water and 2 µL various concentrations of ssRNA/ssDNA target. The fluorescence signal was measured at 30-s intervals at 37 °C using a CFX96 touch real-time PCR system (Bio-Rad, CA, USA).

## Asymmetric CRISPR assay for miRNA detection

The cleavage reaction was conducted in a final volume of 20 µL, including 2 µL 10X NEBuffer r2.1 buffer, 2 µL LbCas12a (1 µM), 0.8 µL full-sized crRNA (1 µM), 0.2 µL each split crRNA (1 µM handle and spacer), 0.4 µL DNA activator (1 µM), 2 µL fluorescence-quencher probe (10 µM), 10.4 µL RNase-free water, and 2 µL various concentrations of target miRNA. The fluorescence signal was measured at 30-s intervals at 37 °C using a CFX96 touch real-time PCR system.

## Asymmetric CRISPR assay for long ssDNA detection

The cleavage reaction was conducted in a final volume of 20 µL, including 2 µL 10X NEBuffer r2.1 buffer, 2 µL LbCas12a (1 µM), 0.8 µL full-sized crRNA (1 µM), 0.2 µL each split crRNA (1 µM handle and spacer), 0.1 µL ssDNA target of split crRNA (1 µM), 1 µL fluorescence-quencher probe (10 µM), 11.7 µL RNase-free water, and 2 µL various concentrations of target ssDNA. The fluorescence signal was measured at 30-s intervals at 37 °C using a CFX96 touch real-time PCR system.

## Gel electrophoresis

For the test using fluorophore-conjugated crRNA, a total of 20 µL of the reaction solution was prepared by mixing 200 nM LbCas12a, 100 nM crRNA (full-sized and split crRNA), and different combinations of 100 nM full-sized crRNA target (full-T) and 50 nM split crRNA target (split-T) in 1 X NEBuffer 2.1 buffer and nuclease-free water. After incubation of the reaction solution at 37 °C for 0 and 30 min, proteinase K was added for 15 min at 45 °C. Then, each reaction solution was resolved on a 10% polyacrylamide gel using 1 x TBE as the running buffer at a constant voltage of 100 V for 100 min. Gels were scanned using a ChemiDoc Imaging System (Bio-Rad).

For the test using fluorophore-conjugated target nucleic acid, a total of 20 µL of the reaction solution was prepared by mixing 100 nM LbCas12a, 40 nM full-sized crRNA, 10 nM split crRNA, and different combinations of 100 nM Cy5-conjugated full-T/Cy3-conjugated non-target and 50 nM FAM-conjugated split-T in 1 X NEBuffer 2.1 buffer and nuclease-free water. After incubation of the reaction solution at 37 °C for 0, 5, 15, and 30 min, each reaction solution was resolved on a 10% polyacrylamide gel using 1 x TBE as the running buffer at a constant voltage of 100 V for 100 min. Gels were scanned using a ChemiDoc Imaging System (Bio-Rad).

## Electrophoretic mobility shift assay

Fluorescence-labeled crRNAs (5′- Cy5 - conjugated full-sized crRNA and 5′- FAM - conjugated split crRNA handle) were mixed in different concentrations. We prepared two groups: (1) Cy5 - full-sized crRNA was fixed at 200 nM and the concentration of the FAM-split crRNA handle varied from 0, 50, 100, 200, and 400 nM. (2) FAM-split crRNA handle was fixed at 200 nM and the concentration of the Cy5 - full-sized crRNA varied from 0, 50, 100, 200, and 400 nM. Next, the prepared crRNA was mixed with 250 nM LbCas12a in 1x NEBuffer 2.1 to prepare a total reaction solution of 20 µL. After incubation for 20 min at 25 °C, the reaction solutions were loaded onto a 5% polyacrylamide gel using 1 x TAE as the running buffer at a constant voltage of 100 V for 100 min[45]. Gels were scanned using the ChemiDoc Imaging System (Bio-Rad).

For the competitive assay, we prepared three groups: (1) 250 nM LbCas12a and 200 nM Cy5-full-sized crRNA, (2) 250 nM LbCas12a, 200 nM Cy5-full-sized crRNA, and 100 nM non-target sequence, and (3) 250 nM LbCas12a, 200 nM Cy5-full-sized crRNA, and 100 nM target sequence. Then, the prepared samples were incubated at 37 °C for 20 min. After incubation, we added different concentrations of FAM-

split crRNA (0, 50, 100, 200, and 400 nM) and incubated at 25 °C for 20 min. Next, the reaction solutions were loaded onto a 5% poly-acrylamide gel using 1 x TAE as the running buffer at a constant voltage of 100 V for 100 min. Gels were scanned using the ChemiDoc Imaging System (Bio-Rad).

## miRNA detection using human plasma samples

Total RNA was extracted from the human plasma samples using the miRNeasy Serum/Plasma Advanced Kit (Qiagen), according to the manufacturer's instructions. The extracted total RNA was diluted 5-fold before testing with the Cas12a assay. Next, the RNA solution was mixed with the asymmetric CRISPR assay reaction components including 1X NEBuffer 2.1 buffer, 100 nM LbCas12a, 40 nM crRNA, 10 nM each split crRNA (handle and spacer), 20 nM DNA activator, and 1 µM fluorescence-quencher probe. The fluorescence signal was measured at 30-s intervals at 37 °C using a CFX96 touch real-time PCR system.

## RNA extraction and RT-qPCR

Total RNA was extracted from the human plasma samples by using the miRNeasy Serum/Plasma Advanced Kit (Qiagen), according to the manufacturer's instructions. cDNA was synthesized from the extracted total RNA using the miRCURY LNA RT Kit (Qiagen). Briefly, a reverse-transcription reaction was performed at 42 °C for 60 min and then inactivated at 95 °C for 5 min. Synthesized cDNA was stored at -4 °C before use. The synthesized cDNA was amplified using the miRCURY LNA SYBR Green PCR Kit and has-miR-19a-3p miRCURY LNA miRNA PCR assay (Qiagen) following the manufacturer's protocol. Total reaction solution contained 5 µL of 2 × SYBR green master mix, 1 µL of PCR primer mix (has-miR-19a-3p miRCURY LNA miRNA PCR assay (Qiagen)), 1 µL of RNase-free water, and 3 µL of cDNA template. PCR was performed with 40 cycles of 95 °C for 10 s and 56 °C for 60 s.

## Reporting summary

Further information on research design is available in the Nature Portfolio Reporting Summary linked to this article.

## Data availability

The main data supporting the findings of this study are available within the article and Supplementary Files. The source data underlying Figs. 2–6 and Supplementary Figs 1–12 are provided as a Source Data file. Specific data P values are also included within the Source Data file. Source data are provided with this paper.

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

## Acknowledgements

We thank Ms. Lorrie Perpetua at the Carole and Ray Neag Comprehensive Cancer Center for clinical samples collection and storage. This work was partially supported by NIH U01CA269147 (to C.L.) and U01AI148306 (to C.L.)

## Author contributions

J.M. designed the study, performed the experiment, and drafted the manuscript. C.L. supervised the concept of the study and reviewed & edited the manuscript.

## Competing interests

The authors declare no competing interests.
