## [Peer Review File · Nature Communications]

Reviewers' Comments:

Reviewer #1:

Remarks to the Author:

In this work, the authors reported a cascade CRISPR assay (or "asymmetric CRISPR assay" according to the authors) to quantitatively detect microRNA (miRNA) without the need for pre-amplification. The assay uses two different crRNAs: a full-size crRNA and a split crRNA, in one pot, with the hypothesis that the full-size crRNA will bind to the target first, and then split crRNA will replace full-size crRNA to generate cascaded signal amplification. The authors also demonstrate that the CRISPR-Cas12a system can detect RNA target, similar to the observation in ref #34. Combining both cascade design and RNA detection capability, the authors finally demonstrated an assay to detect miR-19a in bladder patient samples. The signal amplification strategy is the most interesting part of the work, while detecting RNA by the CRISPR-Cas12a has been more systematically studied in a recent work (ref #34).

Although interesting, one of the major comments is that the mechanism of the cascade amplification is still not clear. The authors attributed the cascade effect to the replacement of full-sized crRNA with the split crRNA, and therefore the initiation of 2nd trans-cleavage reaction. This hypothesis seems problematic, since it is contradictory to the authors' data (Fig 2) that full-size crRNA binds Cas12a more strongly than split crRNA. How exactly can split crRNA replace full size crRNA? Second, if there are truly two-step reactions, then when the full-sized crRNA trans-activates Cas12a, it may not only trans-cleave the F-Q reporter but also the split-target (split-T) as well, which could inhibit the 2nd step reaction. Finally, when looking at the reaction stoichiometry, the asymmetric Cas12a assay seems to have excess Cas12a in most cases (e.g., 100 nM LbCas12a, 40 nM crRNA, 10 nM split crRNA). The amount of Cas12a is higher than the amount of full-size crRNA and split crRNA combined. This raises the question of whether it is a true sequential two-step reaction as the authors proposed, or simply a higher total crRNA concentration in the asymmetric Cas12a assay compared to a conventional Cas12a assay. The cascade mechanism needs to be further explained or validated by new experimental data for future publication.

Other minor comments:

1. There are a few recent studies on trans-cleavage with split crRNA, for instance, *Nucleic Acids Research*, Volume 50, Issue 2, 25 January 2022, Pages 1162–1173 (<https://doi.org/10.1093/nar/gkab1227>). The authors may want to add this citation and modify their corresponding statement a little bit.
2. In Figure 1b, the schematic shows the full-size crRNA and target complex will be released from Cas12a. How is that possible? What is the mechanism?
3. In Figure 2b, the results show that a full-size competitor crRNA cannot replace an already bound full-size crRNA, which is contradictory to the observation in ref #25. Any explanation?
4. Line 104-107: "In the presence of target nucleic acids, the Cas12a/full-sized crRNA complex is specifically activated and initiates the first trans-cleavage reaction, resulting in the cleavage and consumption of full-sized crRNA through Cas12a/crRNA kinetics". This statement seems questionable. How can full-size crRNA get cleaved and consumed in the presence of the target? What is the mechanism?
5. In Figure 3b and c, the gel images are a little bit confusing and difficult to interpret without labeling the markers and bands.
6. In Figure 3b, lane 2 and lane 3 from the left, should there be three bands instead of two, since the mixture contains a full-size crRNA and split crRNA (containing two fragments)?

7. In Figure 3c, lane 2 and lane 3 from the left, when there is no target, why does the band become thinner in lane 3 after 30 min?
8. Figure 3c caption (Line 429-430): "The reduced Cy5 fluorescence signal in the control sample (in the absence of full-T and split-T) indicates pre-crRNA processing." How is pre-crRNA processing relevant here to the assay?
9. Supplementary Figure 1A: Why are the bands in lane 2 and lane 4 from the left very weak, where there should have full-sized crRNA and split crRNA-spacer, respectively?
10. Line 163-166: "Interestingly, when comparing the DNA target detection efficiency in the presence or the absence of the split crRNA and split-T, the signal was further amplified when the split crRNA and split-T were added (Supplementary Figure 2)." This claim does not match well with Supplementary Figure 2. In Supplementary Figure 2, it seems that split crRNA only gives higher signals for low target concentrations but lower signals for higher target concentrations.
11. Results shown in Figures 5b, 5c, and 5d were not discussed in the main text. The authors need to explain the results shown in these figures.
12. Supplementary Figure 8B: why does the control also produce a high fluorescent signal (~2000 a.u.)?
13. Line 235-238: "the fluorescence intensity linearly increased with the logarithm concentration of miR-19a ranging from 1 fM to 10 pM in the presence of split crRNA. Using the equation $y = 645.4x + 10279.5$ ($R^2 = 0.984$), we estimated the LOD to be 196 aM". The authors performed assay with the lowest concentration at 1 fM. Therefore, it may not be appropriate to claim a LOD of 196 aM. Moreover, why is the LOD for miRNA detection much lower than the synthetic targets (LOD ~100 fM, page 8, line 168)?
14. Who provides the patient samples? More information on the sample source is needed.

Reviewer #2:

Remarks to the Author:

Authors Moon and Liu propose a novel method of amplifying CRISPR based RNA detection by combining a split crRNA – guide and a split DNA/RNA target design. The major claims of the work are (i) that by combining full length and split crRNA guides together, sensitivity of nucleic acid detection is improved and (ii) Cas12a2 can recognize RNA targets when a partial DNA is available. Although both of these claims advance the field, the first claim needs more controls to be substantiated and the second claim has already been made elsewhere as a pre-print, although not yet published in a peer-review (<https://www.biorxiv.org/content/10.1101/2023.01.29.525716v1.full>). Additionally, the figures are difficult to understand, making it challenging to analyze the data. Thus, I cannot recommend this work be published at Nature Communications without a significant amount of revision.

Major concerns:

Claim 1 – The split crRNA guide combined with full length provides more sensitivity than existing methods.

The diagram of figure 5 panel A, suggests that depending on the concentration of Cas12a in the assay, at least some Cas12a will be activated by the base ingredients of the assay. Namely the split crRNA and the DNA probe. It is not clear that a control was done to show how much change in fluorescence intensity is generated over time without a target. Such a control is needed for the data presented in figure 5. panels e. and f. that compare with and w/o split. Indeed it is difficult to decipher what substrates were included in the negative controls.

Without the aforementioned control, it is unclear whether the higher sensitivity is due to the split-crRNA method or is an artifact of high background. Additionally, the authors should examine whether the higher sensitivity observed with miR-19a, can also be observed with other RNAs. The authors should also comment on whether there are any secondary or tertiary structures of the

miRNA that would make it a strong or weak target for detection.

Claim 2 – Cas12a2 can detect RNA if a ssDNA that binds to the seed side of the crRNA is available. Although the authors cite the pre-print manuscript that extensively described how Cas12a can be manipulated with seed bound DNA targets to detect RNA, there is very little credit given to the existing work in the text and no comparative analysis. Indeed, more could be done to analyse how different types of DNA probes (e.g. dsDNA, with and with PAMs etc.) or at the very least provide an explanation for why only a ssDNA was used as a probe with the previous work described in context.

Figures –

Figure 1 and throughout the paper the split crRNA should be shown as two distinct RNA molecules in every instance instead of one molecule. By showing it as a single RNA in figure 1 when bound to Cas12a it is very confusing. From what I'm reading in the methods it appears that there is far more Cas12a than the sum of full crRNA and split crRNA (eg. 100 nM LbCas12a, 40 nM crRNA, 10 nM each split crRNA) so it is unclear why the diagram shows the split crRNA causing the release of the full crRNA bound to target. Indeed, it would be more accurate to show some apo, some bound to full-crRNA, and some bound to split-crRNA in the diagram. Additionally, the entire figure seems unnecessary, as it is not really presenting data but rather summarizing the result of the paper. Perhaps a modified version could be used as a visual abstract?

Figure 2. All targets should be clearly indicated in text as DNA or RNA throughout the figures of the paper. The control in the bottom part of panel a should be better described in the figure legend or the panel. It is unclear what and what is not being excluded from the reaction.

Panel c was difficult to understand. The authors should indicate on the figure what crRNA (full or split) is being held constant and what concentration. It may help with understanding if the two FAM – split crRNA images were put together and the two Cy5 – full sized were put together with title indicating cartoons indicating the different crRNAs on the gel.

Figure 3. All bands on the gel should be labeled. The supplemental data that show controls should be referenced in the figure legend, and more controls should be included. For example there are neither controls with non-target sequences nor controls that only have the Cy5-full crRNA with just T or split T to show that the substrates are free of RNase contaminants. Without such controls the claim that the crRNA is getting cleaved can be explained by contaminating RNases removing the Cy5 fluorophore. Indeed, the Sybr-gold stained gels appear to show that most of the full-length crRNA stays hybridized with the target.

The claim that Cas12a cleaves the crRNA is significant, as no other group has shown such an activity to date. To establish this claim the authors should pursue how it is getting cleaved and where on the crRNA it is getting cleaved.

The authors should simply do an EMSA to determine a Kd for the different substrates rather than claim one has higher affinity for the other based on competition.

Figure 5. Fluorescence should be reported when all ingredients including with and w/o split crRNA but without the miRNA. It is unclear what is meant by the 10 pM in the corners of panels b, c, and d.

Figure 6. To make the claim that the split-crRNA is more sensitive to other methods the authors should compare the existing tools such as DETECTR and SHERLOCK in their assay. It would also be helpful to explore a second RNA.

Concerns pointed out by a graduate student reviewer:

1. In line 299 – 300 the authors reference a preprint on Biorxiv, Rananaware et al., 2023 to support the finding that Cas12a can bind RNA. Rananaware et al also report the activation of trans cleavage with split target RNAs and the use of these split targets in RNA diagnostics, major findings also reported in this work. It is therefore concerning that the data of Rananaware is not referenced in more depth. As is, the authors do not sufficiently acknowledge the work of Rananaware et al in this regard.

2. Explanation of full-sized crRNA cleavage does not make sense, could the authors please elaborate.

- a. Can the authors elaborate what is meant by “first trans-cleavage” (line 105 – 106) and “second trans-cleavage” (line 108).
 - b. The authors reference Stella et al., 2018 when making the claim that “[full sized crRNA degradation could be] explained by the release and cleavage of the hybridized structure of the full-sized crRNA and the target nucleic acid, according to previously published Cas12a reaction kinetics.” (lines 149 – 151). Stella et al. showed that 1.) target bound crRNA could be displaced by competitor crRNA and 2.) excessive concentrations of competitor crRNA can be cleaved by trans-cleavage activity. These findings are not the same as what the authors propose here - that the crRNA in complex with the target is cleaved by Cas12a.
 - c. In figure 3c and SF1, the decrease in the crRNA band intensity after 30 minutes (for both full-sized and handle crRNA) indicates that degradation of the crRNA is not necessarily target dependent. For the authors to prove that this degradation is caused by Cas12a, they must show that mutation of the RuvC active site abolishes this crRNA cleavage. Additionally, for the authors to make the claim that the full-sized crRNA must be cleaved before the split crRNA can contribute, they should test the diagnostic ability with a non-hydrolyzable crRNA (phosphorothioate modification) to show activity that matches no-split crRNA added.
 3. In several places throughout this work, the authors “put the cart before the horse” and describe a model for how their methods work before presenting the data that supports the model. For example, in the sentence beginning in line 94 states that full-sized and split crRNAs have different binding affinities while the data that prove this statement are not presented until the paragraph beginning in line 129. This pattern of presenting data out of order makes the rational flow of the work difficult to follow. Data should be presented in such a way as to lead the reader to the same logical conclusion that the authors reach.
 4. In this work, the authors describe a new method for performing RNA diagnostics with asymmetric crRNA activities. However, there are serious problems with the organization of the methods section that would make reproducibility challenging. Some of these problems are listed below:
 - a. Titles of methods sections are not clear enough to unambiguously know to which data within the paper they refer.
 - b. Methods are not sufficiently detailed to be replicated. Key details like incubation times, concentrations etc. are often omitted.
 - c. The specifics of how the asymmetric CRISPR assay is designed are not provided.
 - d. The data presented in figure 2b represent end-point fluorescence measurements, while none of the author provided methods correspond to this experimental set-up. The authors must include the methods of all experiments performed in sufficient detail that they could be replicated.
 - e. The experiment described in lines 189 – 200, there is no clear method describing how these exact experiments were conducted. Specifically, the source of miR-19a is not clear. Were these miR-19a samples in vitro transcribed or ordered or from patient samples?
 5. The authors should use the same language to describe their experimental conditions in both the text and figures. For example, in the text description of the competitive crRNA experiment, the authors use language different from the text in the figure labels. Lack of continuity between the text and figures makes it hard to understand what you are talking about.
- Minor concerns.
1. The term “asymmetric CRISPR assay” is used liberally throughout, but this terminology is never clearly explained.
 2. The authors should cite the LEOPARD CRISPR-based detection method in the introduction.
 3. In line 43, change “developed” to evolved. Developed implies rational design in the emergence of CRISPR-Cas immune systems.
 4. Consider removing or clarifying the sentence beginning “Several” on line 55. How much do the kinds of chemical modifications used in the referenced paper really relate to the work at hand.
 5. Consider rewording the sentence beginning on line 59 to read “Another study reported that a split crRNA containing separated scaffold and spacer RNA can catalyze highly specific cis-cleavage in Cas12a, but did not explore the effect on trans-cleavage.”. The following line could then be removed.
 6. Clarify dsDNA in line 71, as Cas12a does not require a PAM sequence to be activated by ssDNA.
 7. One flaw in existing CRISPR diagnostic methods mentioned in the introduction is the requirement of multiple crRNAs (line 52). Could the authors explain why multiple crRNAs are now beneficial in the authors proposed solution beginning on line 74 (specifically mentioned on line 78).
 8. RE. “... PAM-free detection” (line 83). To make this statement, the authors should include data

and or rational that supports the PAM independence of the method. Since this is not addressed in the body of the work, it should be omitted.

9. In line 91, remove "(DNA or RNA)" as the illustrated set-up in figure 1 does not show how the authors were able to detect RNA. Further modifications were necessary for RNA detection as illustrated in figure 4.

10. In line 132, the authors describe crRNAs used at different ratios, but the exact concentrations used are not mentioned in the text or methods sections.

11. Could the authors explain the use of the phrase cascade signal amplification on lines 140 and 142. Usually, cascade signal amplification refers to mechanisms like phosphorylation cascades or the synthesis of second messengers as seen in type III CRISPR diagnostics (eg. SCOPE). A mechanism of reactivation proposed by the authors does not neatly fit this paradigm and so needs clarification.

12. Lines 166 – 169. Do the authors perhaps mean lower instead of higher when describing their LOD of 100 fM? The authors should also report a.) the LOD determined for Cas12a w/o split crRNA, and b.) the equation used to interpolate the LOD. Additionally, a horizontal line at the mean blank + 3.3σ should be included in SF2 for clarity.

13. Line 178 – 181. The sentence beginning on line 178 should be changed to make the experimental rationale clearer. For example, "probe" is often used to describe the fluorescent reporter used to measure trans-cleavage, though this is apparently not what the authors mean based on figure 4. Furthermore, the phrase "various locations" is extremely ambiguous. Please change the text to make the meaning clearer.

14. Line 180 – 181. Please change the text to clarify what is meant by "when only one type of target is present".

15. Change "to target" in line 192 to "complementary to", for clarity.

16. Can the authors explain why they chose to target different segments of the miR-19a with the RD crRNA vs. DD crRNA? Why not target the same sequence but at the 5' and 3' ends? Can the authors also explain why this particular segment of miR-19a was chosen?

17. The design for detecting RNA presented in figure 5a deviates from the asymmetric CRISPR assay first introduced in figure 1, in that the "DNA-probe" is both complementary to segments of the full-sized crRNA and the split crRNA. The authors should explain their design rationale in the text. Could the authors additionally explain why an independent target for the split crRNA is not used as in figure 1?

18. Line 238. Could the authors compare their amplification free results to published data on amplification free diagnostic potential of Cas12a (eg. Ramachandran and Hyuke 2023).

19. Line 257 – 258. Did the authors include any blinding controls while testing the bladder cancer patient samples against healthy patients, or did the authors know while performing the assay which samples belonged to which group?

20. Are the progress curves presented in Figure 1b experimentally derived? If so, the authors should make that clearer. If not, then these hypothesized outputs should be verified by experimentation.

21. In figure 2c, the authors should present images from the same gel next to each other for easier comparison. Additionally, because two different fluorescent channels are used, the two images could be overlaid for even easier comparison as is common practice when using different fluorophores on nucleic acids (ref).

22. In figure 3b, the authors should identify all the bands present in the gel. This would be made easier if the authors chose to use denaturing Urea-PAGE or FDF-PAGE gels instead of native page. Could the authors explain why they chose not to image for the FAM label on the split crRNA?

Reviewer #3:

Remarks to the Author:

"I co-reviewed this manuscript with one of the reviewers who provided the listed reports. This is part of the Nature Communications initiative to facilitate training in peer review and to provide appropriate recognition for Early Career Researchers who co-review manuscripts."

Reviewer #4:

Remarks to the Author:

Moon et al. reports an asymmetric CRISPR assay for cascade signal amplification detection of nucleic acids by leveraging the asymmetric trans-cleavage behavior of competitive crRNA, with a narrative that positions the work ultimately for de-centralized diagnostics. As a proof-of-concept, the authors used the CRISPR assay to quantitatively detect a microRNA (miRNA) biomarker for liquid biopsy, demonstrating an amplification-free and PAM-free detection approach using CRISPR-Cas12a.

This is an interesting addition to the CRISPR-based sensor toolbox. The fundamental biology involving the full and split crRNA, and how this can be exploited to develop variable affinity to Cas12a is very interesting and may have broader implications. The authors' use of this phenomenon to create a cascade for signal amplification is creative and the reported LODs are impressive. I'm not able to comment on the novelty of these features in the broader CRISPR literature, but for sensing this represents quite a step toward the practical deployment of CRISPR for point-of-need diagnostics.

While I quite like the manuscript, this is a complex project and there are places in the manuscript where additional text and figures may be required to ensure the material is accessible to wide audience. In summary, the manuscript is well-written, and the contents are also organized and, for the most part, clearly demonstrated. I do have some questions and comments for the authors that are summarized below.

General comments

1. During the review process, I suggest that the authors insert the number of lines in the text to make it easier to reference material.
2. Please check minor spelling errors in the main text (e.g. quantitative, etc).
3. Statistical analysis (or a description of the analysis in the figure legend) is missing from the data presented (e.g., Fig. 2, 4, 5 and 6). Notably, we can infer some differences based on the values obtained, but I believe that performing adequate statistical analysis between the conditions tested is required for publication. Further, the graphs in figure three do not have error bars (e.g. 4b, 4c, and 4d). Were these experiments replicated?

Similarly, many graphs in the supplemental information (Fig S3-S9) will need the appropriate statistics details added (# replicates, # experiment, SD/error bars).

4. Figure 1. Authors state "In the presence of target nucleic acids, the Cas12a/full-sized crRNA complex is specifically activated and initiates the first trans-cleavage reaction, resulting in the cleavage and consumption of full-sized crRNA through Cas12a/crRNA kinetics. Subsequently, the split crRNA can replace the cleaved full-sized crRNA and reactivate Cas12a for the second trans-cleavage reaction, which leads to an additional fluorescence signal amplification"

While it seems intuitive that this would be the case. Have authors tested the addition of a high concentration (e.g. 2x) of full-sized crRNA to confirm that the signal from the asymmetric assay is in fact greater? The abundance of the split-T also would seem to factor into better performance of the asymmetric assay and should probably be mentioned.

5. Figure 2c. I'm not sure there is a better way to present the data, but it did take a while to properly interpret this figure.
6. Figure S1. Please consider labeling the bands on the gel with arrow labels along one side.
7. Figure 3b/c. It would be helpful to label the split and full crRNAs on the side of the gel. It would also be useful to add an arrow to indicate the degradation of the split FAM crRNA.
8. Figure 4. While the concept is very interesting and practically useful, the nomenclature of the fragmented target with probe and target segments was at first quite confusing. This is the first

mention of the word "probe" and the concept in general and so deserves some more description. I would also recommend augmenting the schematic to improve understanding (for example, enlarge the images of the probe and/or target with crRNA and Cas12a)

Further, the thin lines and small text in all primary figure schematics makes it difficult to interpret. Please consider increasing line thickness and/or it would be helpful if you can find a way to emphasize the critical details that the readers should be tracking.

The seed region (within 5-10 nt PAM) is referenced as an important potential factor. To make the point more accessible to readers, please consider indicating the PAM sequence (e.g. bold or underline).

9. Figure S3. With the emphasis of the seed region in this data set. Again, marking the seed region and/or the PAM sequence would be helpful.

10. Figure S4. Is the ssDNA probe a critical partner for detection of ssRNA? If so, okay. But the concept needs to be clearly described. Also what are the consequences to practical implementation? Does the probe have tolerance to SNPs if there is mismatch with the biological target?

Page 10. Authors report "For comparison, we designed a miR-19a 3'-target/ssDNA 5'-probe by reversing the targeted locations of the RD crRNA and designed crRNA that binds to ssDNA 5'-probe and miR-19a 3'-target (DD crRNA). Interestingly, they could not initiate the collateral cleavage reaction of Cas12a regardless of the crRNA binding position (Supplementary Figure 4). By contrast, when both target nucleic acids were ssDNA, we observed an efficient cleavage reaction regardless of the binding length (Supplementary Figure 5). Based on this finding, we placed the miRNA at the 5'-target position close to the 3' end of the crRNA."

While data clearly shows that collateral cleavage works if both probe and target are ssDNA (Fig S5) and does not work (for the most part) if the both probe and target are ssDNA and miRNA, respectively (Fig S4). It is not clear how these results guided the authors to place the miRNA at the 5'-target position close to the 3' end of the crRNA? Some additional description of the rationale would be useful.

11. Figure S6. The authors state "The real-time fluorescence signal was measured depending on various concentrations of target miRNA (100 fM ~ 1 nM) and confirmed that a 1 pM level of miRNA could be detected (Supplementary Figure 6)." In looking at the data, it looks more like the lowest detectable miRNA concentration was 100 pM. How are the authors calculating the limit of detection? It is typically calculated as detection above 3x the SD of the negative control. There are no error bars included and so interpretation of the data is difficult.

12. Figure S7. The trans-cleavage fluorescence signal presented is very low (~100 RFU) compared to signal in Figure S6 (e.g. 1×10^4 , 1 nM target) and very close to background. Statistical analysis will be important to determine whether the two conditions are similar or just represent background. Author's should include data from high target concentrations to better elucidate the similarity of detection thresholds (ssDNA vs ssDNA/ssRNA).

13. Figure 5. The description of panel 5a is a helpful summary of how the asymmetric CRISPR assay is rationally designed (pg 10/11). A more detailed description of the design process may help readers onboard the concept into their labs.

14. Figure 5b. Authors state on page 11 "After optimizing the reaction conditions of the split crRNA, DNA probe, and Cas12a in the asymmetric CRISPR assay". It would be important for the readers to understand if these components need to be optimized for each new target sequence or if these represent universal concentrations for the components (with a summary of the conditions selected as optimal).

15. Figure 5 (and S2). The text refers to concentration with femtomolar to nanomolar notation, but the figures use molar with 10^{-x} . It would be good to synchronize these, with a preference to

the use of fM, nM, etc as it is more intuitive for readers.

16. Figure 5 ef. The authors state "Interestingly, in the high-concentration target miRNA condition, there was no significant difference compared with the CRISPR detection without split crRNA; however, a notable difference in fluorescence signal emerged in the low-concentration target condition." While I agree that the signal difference is less at higher concentrations, there is a strong statistical difference between the two modes of detection at higher concentrations in Figure 5f. Also, the author's present the data as delta FL intensity. Please define the calculation, for instance the change in fluorescence over what period of time?

Authors state "As shown in Figure 5f, the overall fluorescence signal was enhanced by the addition of split crRNA, and the detection sensitivity was significantly improved compared with that without split crRNA." This should read, the "change in fluorescence signal" to reflect the data presented.

Further, it would be important to include the raw data (RFUs) of both modes of detection in the supplemental data.

17. Figure S8. As the author's highlight, the asymmetric CRISPR assay appears to provide a dose dependent response. However, statistical analysis (I assume SD is indicated by line thickness, but this is not indicated), it is difficult to determine with what level of confidence detection can be assigned. Further, with the RT-qPCR (S9) as contrast, the importance of dynamic range of detection is also key. The difference between control (nc) and the lower concentration target containing samples is quite small.

It is worth noting that while the reported assay is isothermal and does not require pre-amplification, the current mode of signal generation would require a plate reader to discriminate between positive and negative results in the clinically relevant range of target concentration (femtomolar and below). This is no problem in a proof-of-concept report, but this caveat should be mentioned in the discussion as a challenge that would need to be addressed for deployment as a practical point-of-need diagnostic.

18. The authors claim their methodology to be "simple/rapid", but this is not the case here in my opinion. In fact, making the CRISPR-based assays requires substantial skills, laboratory infrastructure, expensive equipment (e.g. plate reader, real-time PCR instrument), electricity, etc. Therefore, the terms should be used very carefully.

19. Analytical sensitivity experiment.

Authors state "In particular, the fluorescence intensity linearly increased with the logarithm concentration of miR-19a ranging from 1 fM to 10 pM in the presence of split crRNA. Using the equation $y = 645.4x + 10279.5$ ($R^2 = 0.984$), we estimated the LOD to be 196 aM, which is 5,000 times lower than the CRISPR detection in the absence of split crRNA, which has a LOD of 1 pM"

The authors describe that the developed CRISPR assay was able to detect concentrations around aM. Also, the authors claim that asymmetric CRISPR has a sensitivity of 5,000 times lower than the CRISPR detection in the absence of split crRNA. However, this result is not clear (Fig. S6 and Fig. S8). These low concentrations (e.g. 1 fM, 10 pM) show a low signal close to the control.

The theoretical LOD can not be inferred by an equation, but rather needs to be demonstrated by testing the related target concentrations empirically.

How many times was this experiment repeated? In general, it is common to observe failure in the detection of very low concentrations of nucleic acid.

Have the authors had experience with false positive results?

I suggest authors use probit analysis to determine LOD. This analysis may help answer some of my concerns about LOD.

20. Figure 6. It is not clear how the authors correlated the asymmetric CRISPR and RT-qPCR results in figure 6c,d. How did the authors normalize the data from both techniques? This information must be clear in the manuscript.

21. Figure 6 - I have a concern about two patient samples used in this study. Samples #5 and #8 showed very low signal intensity. Could the authors do statistical analysis comparing these samples with the healthy (negative) samples?

22. Additional discussion is needed - The authors provided a concise and brief discussion of the work, which was well-written. However, with the author's emphasis on de-centralized diagnostic in mind, what would be the cost to perform one reaction using the reported methodology? In terms of application and distribution to low-income countries and many point-of-need applications, this is a critical factor. I believe that the readers would benefit from a calculation for the cost of one reaction, for example. It may also be interesting to address potential future directions in the manuscript.

23. Additional discussion is needed - I would like to see more content regarding the advantages and barriers to producing and distributing CRISPR-based technologies in low-resource settings (e.g., remote areas, rural cities, etc). What would be the next steps to make the methodology used in the field? What are the bottlenecks to promote decentralization? This was exemplified by the fact that there is no CRISPR-based system being used in reference laboratories for the diagnostics of infectious diseases.

24. What are the limitations of this study? Possible drawbacks of this approach are not clearly spelled out and should be mentioned in the Discussion.

25. Materials and Methods - I have a concern about the methods section. Overall, the authors provided important details for the execution of the methodology. I understand that the methods should be condensed to make it easier to read, but I am in favor of transparent science. In my opinion, I suggest the authors provide more details of the methodology in a way that allows the reproduction of the results. These details can be inserted in the supplementary material. Please see supplementary material from the manuscript "Sensitive fluorescence detection of SARS-CoV-2 RNA in clinical samples via one-pot isothermal ligation and transcription" as an example: <https://www.nature.com/articles/s41551-020-00617-5>

26. Patient sample collection - The authors used patient samples in the study and did not provide information how the samples were collected. Please insert these details.

27. Discussion - The authors bring some sentences to the discussion that seem to overvalue the reported data in the manuscript. For example, authors highlight that the reported methodology "could have important clinical applications in early cancer diagnostics and infectious disease detection". I suggest authors to be careful with this type of claim, since only one target was used in the current study, nor were any patient trials performed using a large number of clinical samples.

Comments in reference to specific lines in the text or figures

1. Figure 2c and related text. The concentration of the FAM-split crRNA should be inserted in the main figure.
2. Figure 3. Please provide the ladder information for each gel.
3. RNA extraction and RT-qPCR. Please provide details about the RT-qPCR assay (e.g. volume of reaction, primer concentration, instrument used, oligonucleotide sequences, etc). This will allow trainees to replicate your results. Did the authors use any housekeeping gene in the analysis of the relative miRNA expression?
4. Methods - Please provide details about the all kits used in the study (e.g., ID catalog).
5. Figure 4b. The legend makes it difficult to see the amplification curve. Please edit.
6. Supplementary Figure 2b. Please use the concentration of ssDNA (10 fM ~ 100 pM) instead e.g.

-9,-10,-11, etc. This minor change will put the graph more easy for the reader.

Reviewer #5:

Remarks to the Author:

I co-reviewed this manuscript with one of the reviewers who provided the listed reports. This is part of the Nature Communications initiative to facilitate training in peer review and to provide appropriate recognition for Early Career Researchers who co-review manuscripts

Response to Reviewer 1's Comments:

We appreciate to reviewer 1 for taking his/her valuable time to comment. Below, we have presented reviewer 1's comment in italics, and the corresponding responses are marked in blue font.

Reviewer #1 (Remarks to the Author):

In this work, the authors reported a cascade CRISPR assay (or “asymmetric CRISPR assay” according to the authors) to quantitatively detect microRNA (miRNA) without the need for pre-amplification. The assay uses two different crRNAs: a full-size crRNA and a split crRNA, in one pot, with the hypothesis that the full-size crRNA will bind to the target first, and then split crRNA will replace full-size crRNA to generate cascaded signal amplification. The authors also demonstrate that the CRISPR-Cas12a system can detect RNA target, similar to the observation in ref #34. Combining both cascade design and RNA detection capability, the authors finally demonstrated an assay to detect miR-19a in bladder patient samples. The signal amplification strategy is the most interesting part of the work, while detecting RNA by the CRISPR-Cas12a has been more systematically studied in a recent work (ref #34).

Although interesting, one of the major comments is that the mechanism of the cascade amplification is still not clear. The authors attributed the cascade effort to the replacement of full-sized crRNA with the split crRNA, and therefore the initiation of 2nd trans-cleavage reaction. This hypothesis seems problematic, since it is contradictory to the authors' data (Fig 2) that full-size crRNA binds Cas12a more strongly than split crRNA. How exactly can split crRNA replace full size crRNA?

Response: Thank you for your positive comments and suggestions.

Yes, as shown in **Figure 2c**, we could observe that the full-sized crRNA can bind to Cas12a more strongly than split crRNA. In other words, when full-sized crRNA and split crRNA were simultaneously mixed with Cas12a, the affinity binding reaction with Cas12a by the full-sized crRNA proceeded dominantly while the reaction by the split crRNA was inhibited by the full-sized crRNA. In the presence of full-sized crRNA, split crRNA could not activate Cas12a enzyme despite the presence of split-T (target of split crRNA) (**Figure 3b, lanes 6, 7**). However, we could observe that the split crRNA could activate Cas12a enzyme when full-T (the target of the full-sized crRNA) was added (**Figure 3b, lane 9**).

From these results, we thought that the full-sized crRNA reaction acted as a trigger process to induce the second split crRNA reaction. To explain this phenomenon, we hypothesized that the full-sized crRNA could be replaced with split crRNA based on the previous study (**Reference #26**; Stella., et al., 2018). In **Reference #26**, they have proposed a conformational resetting mechanism in which the new crRNA can replace the crRNA-DNA hybrid or the cleaved R-loop after CRISPR-Cas12a cleavage reaction. In addition, we have further validated the full-sized crRNA replacement by split crRNA through electrophoretic mobility shift assay (**Figure 3c**).

Thus, based on our experimental results (**Figure 3**) and **Reference #26**, in the presence of full-T (the target of the full-sized crRNA), we thought that the full-sized crRNA can be replaced by split crRNA, initiating a cascaded CRISPR signal amplification. We have included these additional experimental data and discussion in the revised manuscript (**Lines 147-171**).

Second, if there are truly two-step reactions, then when the full-sized crRNA trans-activates Cas12a, it may not only trans-cleave the F-Q reporter but also the split-target (split-T) as well, which could inhibit the 2nd step reaction.

Response: Yes, the split target (split-T) can be cleaved by the Cas12a/full-sized crRNA trans-cleavage reaction. However, we used a relatively high concentration of split-T in our asymmetric CRISPR assay. In addition, we expect that the split-T cleavage reaction by Cas12a/full-sized crRNA could be reduced due to the formation of a hybrid structure of split-T and split crRNA spacer.

Finally, when looking at the reaction stoichiometry, the asymmetric Cas12a assay seems to have excess Cas12a in most cases (e.g., 100 nM LbCas12a, 40 nM crRNA, 10 nM split crRNA). The amount of Cas12a is higher than the amount of full-size crRNA and split crRNA combined. This raises the question of whether it is a true sequential two-step reaction as the authors proposed, or simply a higher total crRNA concentration in the asymmetric Cas12a assay compared to a conventional Cas12a assay. The cascade mechanism needs to be further explained or validated by new experimental data for future publication.

Response: Thank you for your suggestion. To explain it more clearly, we performed additional experiment by using 5'-Cy5-conjugated full-T (full-sized crRNA's target) and 5'-FAM-conjugated split-T (split crRNA's target) under the same reaction conditions (e.g., 100 nM LbCas12a, 40 nM full-sized crRNA, and 10 nM split crRNAs). We incubated samples for 0, 5, 15, and 30 min respectively, and then run the PAGE gel.

From the results (**Figure 3a**), we could see the full-sized crRNA reaction can occur regardless of split crRNAs (**Lanes 2 – 5**). However, in the case of split crRNA, the cleavage reaction cannot occur in the presence of full-sized crRNA (**Lanes 6 – 9**). Although an excess of LbCas12a was used compared to the sum of full-sized crRNA and split crRNA, an inhibition effect against split crRNA could be confirmed in the presence of full-sized crRNA. We also demonstrated that split crRNA cleavage reaction can occur in the absence of full-sized crRNA. (**Supplementary Figure 1a**)

In addition, we also compared the cleavage reaction using a 5'-Cy3 conjugated non-target sequence (non-complementary sequence to the full-sized crRNA). As shown in **Supplementary Figure 1b**, the full-sized crRNA remained inactive in the presence of the non-target sequence, and the cleavage reaction of the split crRNA was also inhibited even in the presence of split-T due to the inactive full-sized crRNA.

Further, we tried to show the cascade reaction using a competition assay, following the experimental method in **Reference #26** (Stella., et al., 2018). We incubated Cas12a and Cy5-full-sized crRNA for 20 min and then treated FAM-split crRNA with different concentrations. At the same time, we also incubated the sample including Cas12a, Cy5-full-sized crRNA, and full-T for 20 min and then added FAM-split crRNA.

As shown in **Fig. 3c**, it could be confirmed that Cy5-full-sized crRNA continues to bind to Cas12a regardless of FAM-split crRNA in the absence of full-T. On contrary, in the presence of full-T, the binding of full-sized crRNA is reduced, and FAM-split crRNA can replace full-sized crRNA to bind to Cas12a. This indicates that the split crRNA, which is initially inhibited by the full-sized crRNA, can reset Cas12a by replacing the full-sized crRNA after the full-sized crRNA is activated by full-T. In other words, from the above results, we confirmed that after the first CRISPR reaction induced by full-sized crRNA and full-T has occurred, the second CRISPR reaction induced by split crRNA and split-T proceeds sequentially. We have included these additional experimental data and discussion in the revised manuscript (**Lines 138-171**).

Other minor comments:

1. There are a few recent studies on trans-cleavage with split crRNA, for instance, *Nucleic Acids Research*, Volume 50, Issue 2, 25 January 2022, Pages 1162–1173 (<https://doi.org/10.1093/nar/qkab1227>). The authors may want to add this citation and modify their corresponding statement a little bit.

Response: We have cited this reference (current **Reference #19**, previous **Reference #18**) in our manuscript.

2. In Figure 1b, the schematic shows the full-size crRNA and target complex will be released from Cas12a. How is that possible? What is the mechanism?

Response: To better explain this, we referred to a previous study of **Reference #26** (Stella., et al., 2018) as mentioned above. In this previous study, they showed that the cleaved R-loop can be displaced in the Cas12a enzyme by a new crRNA molecule. They proposed a mechanism for resetting and generating another RNA-guided endonuclease by exchanging the crRNA-DNA hybrid or the R-loop after cleavage by another crRNA molecule. In addition, our experimental results (**Figure 3c**) have further confirmed it.

Based on the premise above, we hypothesize that after the target recognition and cleavage reaction of the full-sized crRNA, it is replaced by a new crRNA, split crRNA, followed by the simultaneous release of the full-sized crRNA and target hybrid structure. We have added more explanations in the revised manuscript (**Lines 163-171**).

3. In Figure 2b, the results show that a full-size competitor crRNA cannot replace an already bound full-size crRNA, which is contradictory to the observation in ref #25. Any explanation?

Response: In **Figure 2b**, we used two different full-sized crRNAs: i) one is target-specific full-sized crRNA, and ii) another is a competitor crRNA that is unable to bind the target sequence. In our testing, we used two full-sized crRNAs and one target sequence for the full-sized crRNA. In other words, there is no target sequence for the competitor crRNA.

4. Line 104-107: "In the presence of target nucleic acids, the Cas12a/full-sized crRNA complex is specifically activated and initiates the first trans-cleavage reaction, resulting in the cleavage and consumption of full-sized crRNA through Cas12a/crRNA kinetics". This statement seems questionable. How can full-size crRNA get cleaved and consumed in the presence of the target? What is the mechanism?

Response: As mentioned above, we referred to a previous study of **Reference #26** (Stella., et al., 2018) to explain the mechanism. In this previous study, they showed that the cleaved R-loop can be displaced in the Cas12a enzyme by a new crRNA molecule. They proposed a mechanism for resetting and generating another RNA-guided endonuclease by exchanging the crRNA-DNA hybrid or the R-loop after cleavage by another crRNA molecule. In addition, our experimental results (**Figure 3c**) have further confirmed it. Also, it had been reported that the 3'- and 5'-ends of crRNA are trimmed after the reaction between the target and Cas12a/crRNA according to previous literature¹. We have added more explanations in the revised manuscript (**Lines 163-171**).

5. In Figure 3b and c, the gel images are a little bit confusing and difficult to interpret without labeling the markers and bands.

Response: Following the reviewer's comment, we have modified **Figure 3** to include labels for the markers and bands. To avoid any confusion and facilitate the interpretation, we have further merged gel images measured with different fluorescence filters into single gel image.

6. In Figure 3b, lane 2 and lane 3 from the left, should there be three bands instead of two, since the mixture contains a full-size crRNA and split crRNA (containing two fragments)?

Response: This is thought to be due to overlapping bands, as the sizes of the two split crRNAs are similar (21 bp each).

7. In Figure 3c, lane 2 and lane 3 from the left, when there is no target, why does the band become thinner in lane 3 after 30 min?

Response: We have described this as pre-crRNA processing. It was reported that the Cas12a protein harbors an intrinsic endoribonuclease activity that enables it to cleave pre-crRNA directly upstream of the crRNA pseudoknot, different from other CRISPR-Cas systems that rely on standalone ribonucleases for pre-crRNA processing.²

In another previous study¹, they have also showed similar gel results that the modified crRNA is trimmed by LbCas12a on its 5'-end in the absence of the activator (the target).

8. Figure 3c caption (Line 429-430): "The reduced Cy5 fluorescence signal in the control sample (in the absence of full-T and split-T) indicates pre-crRNA processing." How is pre-crRNA processing relevant here to the assay?

Response: We included this sentence to better explain why the bands corresponding to the Cy5-full-sized crRNA were reduced despite being without a target sequence (full-T).

9. Supplementary Figure 1A: Why are the bands in lane 2 and lane 4 from the left very weak, where there should have full-sized crRNA and split crRNA-spacer, respectively?

Response: This is thought to be because each full-sized crRNA and split crRNA is an RNA sample. The size of the full-sized crRNA is 42 bp and the split crRNA spacer is 21 bp. To make it clearer, we have performed additional experiment by using Cy5-full-sized crRNA and FAM-split crRNA handle, and merged gel images measured using different fluorescence filters into a single gel image as shown in **Supplementary Figure 2a**.

10. Line 163-166: "Interestingly, when comparing the DNA target detection efficiency in the presence or the absence of the split crRNA and split-T, the signal was further amplified when the split crRNA and split-T were added (Supplementary Figure 2)." This claim does not match well with Supplementary Figure 2. In Supplementary Figure 2, it seems that split crRNA only gives higher signals for low target concentrations but lower signals for higher target concentrations.

Response: Based on the reviewer's comment, we have re-written the claim in the revised manuscript (**Lines 181-184**).

11. Results shown in Figures 5b, 5c, and 5d were not discussed in the main text. The authors need to explain the results shown in these figures.

Response: As suggested by reviewer, we have added more discussion on Figures 5b, c, and d in the revised manuscript (**Lines 245-258**).

12. Supplementary Figure 8B: why does the control also produce a high fluorescent signal (~2000 a.u.)?

Response: This is because the control sample also contains full-sized crRNA, split crRNA, and DNA activator (split crRNA target). Although the *trans*-cleavage reaction by split crRNA and DNA activator is significantly inhibited by the full-sized crRNA, the fluorescence signal is possibly induced since the reaction cannot be fully inhibited.

13. Line 235-238: "the fluorescence intensity linearly increased with the logarithm concentration of miR-19a ranging from 1 fM to 10 pM in the presence of split crRNA. Using the equation $y = 645.4x + 10279.5$ ($R^2 = 0.984$), we estimated the LOD to be 196 aM". The authors performed assay with the lowest concentration at 1 fM. Therefore, it may not be appropriate to claim a LOD of 196 aM. Moreover, why is the LOD for miRNA detection much lower than the synthetic targets (LOD ~100 fM, page 8, line 168)?

Response: We initially used a linear regression curve to calculate the LOD as described in previous studies.^{3, 4, 5} The ICH (International Conference on Harmonization) guideline Q2(R1) on validation of analytical procedures also provided the LOD calculation equation, $LOD = 3.3 \times \sigma / m$ (σ is the standard deviation of the response and m is the slope of the calibration curve).⁶ As suggested by reviewer, we have claimed the LOD based on the lowest target concentration we could detect.

The miRNA target used in **Figure 5** is also synthetic RNA target.

14. Who provides the patient samples? More information on the sample source is needed.

Response: The patient samples were provided by the Carole and Ray Neag Comprehensive Cancer Center with a protocol approved by the ethics committee at the University of Connecticut Health Center (IRB #08-310-1). We added more details in the revised manuscript (**Lines 374-377; 573-574**).

Response to Reviewer 2's Comments:

We appreciate reviewer 2 for taking his/her valuable time to comment. Below, we have presented reviewer 2's comment in italics, and the corresponding responses are marked in blue font.

Reviewer #2 (Remarks to the Author):

Authors Moon and Liu propose a novel method of amplifying CRISPR based RNA detection by combining a split crRNA – guide and a split DNA/RNA target design. The major claims of the work are (i) that by combining full length and split crRNA guides together, sensitivity of nucleic acid detection is improved and (ii) Cas12a2 can recognize RNA targets when a partial DNA is available. Although both of these claims advance the field, the first claim needs more controls to be substantiated and the second claim has already been made elsewhere as a pre-print, although not yet published in a peer-review (<https://www.biorxiv.org/content/10.1101/2023.01.29.525716v1.full>). Additionally, the figures are difficult to understand, making it challenging to analyze the data. Thus, I cannot recommend this work be published at Nature Communications without a significant amount of revision.

Major concerns:

Claim 1 – The split crRNA guide combined with full length provides more sensitivity than existing methods. The diagram of figure 5 panel A, suggests that depending on the concentration of Cas12a in the assay, at least some Cas12a will be activated by the base ingredients of the assay. Namely the split crRNA and the DNA probe. It is not clear that a control was done to show how much change in fluorescence intensity is generated over time without a target. Such a control is needed for the data presented in figure 5. panels e. and f. that compare with and w/o split. Indeed, it is difficult to decipher what substrates were included in the negative controls. Without the aforementioned control, it is unclear whether the higher sensitivity is due to the split-crRNA method or is an artifact of high background.

Response: Thank you for your comments. All data in **Figure 5** were presented as Δ FL intensity, which means the fluorescence signal differences between the target RNA sample and the control sample (without target RNA).

For example, in the case of 'with split crRNA group' in **Figures 5f** and **5g**, both the target RNA sample and control sample were first prepared to include full-sized crRNA, split crRNA, and DNA probe (currently re-termed as DNA activator). Then, each fluorescence signal was measured in real-time, and the Δ FL intensity was obtained by subtracting the fluorescence signal of the control sample from the fluorescence signal of the target sample.

We have also provided the raw fluorescence graph of **Figure 5g** in the **Supplementary Figure 9**. To make it clearer, we have added more details in the revised manuscript (**Lines 266-269**).

Additionally, the authors should examine whether the higher sensitivity observed with miR-19a, can also be observed with other RNAs. The authors should also comment on whether there are any secondary or tertiary structures of the miRNA that would make it a strong or weak target for detection.

Response: Following the reviewer's comment, we have tested other miRNA (e.g., let-7a miRNA) to confirm that this assay can be applied to various types of miRNAs as shown in **Supplementary Figure 10**. We utilized the same DNA probe used for miR-19a detection and designed a full-sized crRNA that binds both the let-7a miRNA target and the DNA probe in the same way we designed

the crRNA for miR-19a detection. Also, since the same DNA probe was used, the identical split crRNA could be used.

As suggested by reviewer, we have added some comments on the effect of secondary or tertiary structures of the target sequence in the revised manuscript (**Lines 255-257**).

Claim 2 – Cas12a2 can detect RNA if a ssDNA that binds to the seed side of the crRNA is available. Although the authors cite the pre-print manuscript that extensively described how Cas12a can be manipulated with seed bound DNA targets to detect RNA, there is very little credit given to the existing work in the text and no comparative analysis. Indeed, more could be done to analyze how different types of DNA probes (e.g. dsDNA, with and with PAMs etc.) or at the very least provide an explanation for why only a ssDNA was used as a probe with the previous work described in context.

Response: As suggested by reviewer, we have added more discussion and explanation on pre-print manuscript ⁷ for RNA detection using Cas12a in the revised manuscript (**Lines 204-207**).

Figures

Figure 1 and throughout the paper the split crRNA should be shown as two distinct RNA molecules in every instance instead of one molecule. By showing it as a single RNA in figure 1 when bound to Cas12a it is very confusing. From what I'm reading in the methods it appears that there is far more Cas12a than the sum of full crRNA and split crRNA (eg. 100 nM LbCas12a, 40 nM crRNA, 10 nM each split crRNA) so it is unclear why the diagram shows the split crRNA causing the release of the full crRNA bound to target. Indeed, it would be more accurate to show some apo, some bound to full-crRNA, and some bound to split-crRNA in the diagram. Additionally, the entire figure seems unnecessary, as it is not really presenting data but rather summarizing the result of the paper. Perhaps a modified version could be used as a visual abstract?

Response: Thank you for your suggestion. As suggested by reviewer, we have modified the split crRNA to show it as two distinct RNA molecules. Also, we have modified the **Figure 1** to make it clearer.

To explain the sequential reaction more clearly, we have done additional testing experiment by using 5'-Cy5-conjugated full-T (full-sized crRNA's target) and 5'-FAM-conjugated split-T (split crRNA's target) under the same reaction conditions including 100 nM LbCas12a, 40 nM full-sized crRNA, and 10 nM split crRNAs. We incubated samples for 0, 5, 15, and 30 min respectively, and then run the PAGE gel.

As shown in **Figure 3a**, we could see the full-sized crRNA reaction can occur regardless of split crRNAs (**Lanes 2 – 5**). However, in the case of split crRNA, the cleavage reaction cannot occur in the presence of full-sized crRNA (**Lanes 6 – 9**). Although an excess of LbCas12a was used compared to the sum of full-sized crRNA and split crRNA, an inhibition effect against split crRNA could be confirmed in the presence of full-sized crRNA. We further confirmed that split crRNA cleavage reaction can occur in the absence of full-sized crRNA (**Supplementary Figure 1a**).

We also compared the cleavage reaction using a 5'-Cy3 conjugated non-target sequence (non-complementary sequence to the full-sized crRNA). As shown in **Supplementary Figure 1b**, the full-sized crRNA remained inactive in the non-target sequence, and the cleavage reaction of the split crRNA was also inhibited even in the presence of split-T due to the inactive full-sized crRNA.

Further, we tried to show the cascade reaction using a competition assay, following the experimental method in **Reference #26** (Stella., et al., 2018). We incubated Cas12a and Cy5-full-

sized crRNA for 20 min and then treated FAM-split crRNA with different concentrations. At the same time, we also incubated the sample including Cas12a, Cy5-full-sized crRNA, and full-T for 20 min and then added FAM-split crRNA.

As shown in **Figure 3c**, it could be confirmed that Cy5-full-sized crRNA continues to bind to Cas12a regardless of FAM-split crRNA in the absence of full-T. On contrary, in the presence of full-T, the binding of full-sized crRNA is reduced, and FAM-split crRNA can replace full-sized crRNA to bind to Cas12a. This indicates that the split crRNA, which is initially inhibited by the full-sized crRNA, can reset Cas12a by replacing the full-sized crRNA after the full-sized crRNA is activated by full-T. In other words, from the above results, we confirmed that after the first CRISPR reaction induced by full-sized crRNA and full-T has occurred, the second CRISPR reaction induced by split crRNA and split-T proceeds sequentially. We have included these additional experimental data and discussion in the revised manuscript (**Lines 138-171**).

Figure 2. All targets should be clearly indicated in text as DNA or RNA throughout the figures of the paper. The control in the bottom part of panel a should be better described in the figure legend or the panel. It is unclear what and what is not being excluded from the reaction. Panel c was difficult to understand. The authors should indicate on the figure what crRNA (full or split) is being held constant and what concentration. It may help with understanding if the two FAM – split crRNA images were put together and the two Cy5 – full sized were put together with title indicating cartoons indicating the different crRNAs on the gel.

Response: Thank you for your suggestion. We have indicated the targets in the text and summarized the sequences of all DNA and RNA in **Supplementary Table 3**.

Figure 2a is a schematic diagram showing how the cleavage reaction differs depending on the type of target-specific crRNA (full-sized crRNA and split crRNA) in the presence of competitor crRNA (full-sized crRNA and split crRNA). In other words, the bottom part of **Figure 2** panel a is not a control, but a schematic diagram showing that target-specific split crRNA cannot proceed with the reaction despite the presence of the target DNA when full-sized competitor crRNA is present. The corresponding results of **Figure 2a** has been shown in **Figure 2b**. To make it clearer, we have added more details in the figure legend in the revised manuscript.

As suggested by reviewer, we have revised **Figure 2c** by merging gel images measured by Cy5- and Alexa488-fluorescence filters into a single image.

Figure 3. All bands on the gel should be labeled. The supplemental data that show controls should be referenced in the figure legend, and more controls should be included. For example, there are neither controls with non-target sequences nor controls that only have the Cy5-full crRNA with just T or split T to show that the substrates are free of RNase contaminants. Without such controls the claim that the crRNA is getting cleaved can be explained by contaminating RNases removing the Cy5 fluorophore. Indeed, the Sybr-gold stained gels appear to show that most of the full-length crRNA stays hybridized with the target.

The claim that Cas12a cleaves the crRNA is significant, as no other group has shown such an activity to date. To establish this claim the authors should pursue how it is getting cleaved and where on the crRNA it is getting cleaved.

The authors should simply do an EMSA to determine a Kd for the different substrates rather than claim one has higher affinity for the other based on competition.

Response: Following the reviewer's comment, we have tested more control samples as shown in **Figure 3b**, **Supplementary Figures 1** and **2**. An identical removal of Cy5 fluorophore from the 5' end of the full-sized crRNA was observed. We think this was not caused by an RNase contaminant, but the 5'-end crRNA trimming process by Cas12a¹. From the gel result of **Figure 3b (Lanes 5 and 9)**, we could see the generation of Cy5-containing full-sized crRNA fragment after 30 min reaction in the presence of target (full-T).

Although the electrophoretic mobility shift analysis (EMSA) can provide quantitative data for determining binding stoichiometries, affinities, and kinetics under appropriate conditions, they are often used for qualitative purposes. The assay can differ for each purpose and a large number of variants have been reported.⁸ For example, a competitive EMSA can be used to evaluate the specificity of protein binding activity to DNA using the same sequence but unlabeled DNA fragment,⁹ and can be used to determine the specific inhibition effect of competitor DNA on protein/target DNA interaction.¹⁰

In this study, through the EMSA results, we confirmed that each full-sized crRNA and split crRNA can bind to Cas12a, however, when the two crRNA are mixed, they show different binding patterns to Cas12a (**Figures 2c** and **3c**). As the concentration of full-sized crRNA increases, the binding between split crRNA and Cas12a decreases, suggesting that both crRNAs competitively bind to the same region at Cas12a.

However, the binding of full-sized crRNA to Cas12a was not affected even when the concentration of split crRNA increased (**Figures 2c** and **3c**). Based on this, we explained that full-sized crRNA had higher binding affinity to Cas12a than split crRNA, preventing the split crRNA from binding to Cas12a. We have included additional experimental data and added more discussions in the revised manuscript (**Lines 138-171**).

Figure 5. Fluorescence should be reported when all ingredients including with and w/o split crRNA but without the miRNA. It is unclear what is meant by the 10 pM in the corners of panels b, c, and d.

Response: As with the reviewer's comments, we have tested the control sample (without the miRNA). The fluorescence signal differences between the control and target samples ($\Delta\text{FL} = F_{\text{Target miRNA}} - F_{\text{Control}}$) have been presented in the **Figure 5**. 10 pM in the corners of panels b, c, and d indicate the target miRNA concentration and ΔFL intensity showed $F_{\text{Target miRNA}} - F_{\text{Control}}$, respectively. To make it clearer, we have also added more details in the figure legend.

Figure 6. To make the claim that the split-crRNA is more sensitive to other methods the authors should compare the existing tools such as DETECTR and SHERLOCK in their assay. It would also be helpful to explore a second RNA.

Response: Following the reviewer's comment, we have compared our assay with other previous studies in **Supplementary Table 1**. Also, we have tested other RNA (e.g., let-7a) as shown in **Supplementary Figure 10**.

Concerns pointed out by a graduate student reviewer:

1. In line 299 – 300 the authors reference a preprint on Biorxiv, Rananaware et al., 2023 to support the finding that Cas12a can bind RNA. Rananaware et al also report the activation of trans cleavage with split target RNAs and the use of these split targets in RNA diagnostics, major findings also reported in this work. It is therefore concerning that the data of Rananaware is not

referenced in more depth. As is, the authors do not sufficiently acknowledge the work of Rananaware et al in this regard.

Response: Thank you for your suggestion. We have revised the manuscript to include more details and discussion on pre-print manuscript⁷ in the revised manuscript (**Lines 204-207**).

2. Explanation of full-sized crRNA cleavage does not make sense, could the authors please elaborate.

a. Can the authors elaborate what is meant by “first trans-cleavage” (line 105 – 106) and “second trans-cleavage” (line 108).

b. The authors reference Stella et al., 2018 when making the claim that “[full sized crRNA degradation could be] explained by the release and cleavage of the hybridized structure of the full-sized crRNA and the target nucleic acid, according to previously published Cas12a reaction kinetics.” (lines 149 – 151). Stella et al. showed that 1.) target bound crRNA could be displaced by competitor crRNA and 2.) excessive concentrations of competitor crRNA can be cleaved by trans-cleavage activity. These findings are not the same as what the authors propose here - that the crRNA in complex with the target is cleaved by Cas12a.

c. In figure 3c and SF1, the decrease in the crRNA band intensity after 30 minutes (for both full-sized and handle crRNA) indicates that degradation of the crRNA is not necessarily target dependent. For the authors to prove that this degradation is caused by Cas12a, they must show that mutation of the RuvC active site abolishes this crRNA cleavage. Additionally, for the authors to make the claim that the full-sized crRNA must be cleaved before the split crRNA can contribute, they should test the diagnostic ability with a non-hydrolyzable crRNA (phosphorothioate modification) to show activity that matches no-split crRNA added.

Response: In this manuscript, we reported a cascade signal amplification based on a competitive reaction between full-sized crRNA and split crRNA. Specifically, this involves sequential reactions from full-sized crRNA reactions to split crRNA. We referred to the *trans*-cleavage reaction triggered by full-sized crRNA as the ‘first *trans*-cleavage reaction’ and the *trans*-cleavage reaction caused by split crRNA was referred to as the ‘second *trans*-cleavage reaction’.

As with the reviewer’s comment, one of the main claims of the previous study (**Reference # 26**, Stella et al., 2018) is that the cleaved R-loop can be replaced with a new crRNA molecule in the enzyme (Cas12a). In this literature (**Reference # 26**), they proposed a mechanism for resetting and generating another RNA-guided endonuclease by exchanging the crRNA-DNA hybrid or the R-loop after cleavage by another crRNA molecule.

Further, we tried to show the cascade reaction using a competition assay, following the experimental method in **Reference #26**. We incubated Cas12a and Cy5-full-sized crRNA for 20 min and then treated FAM-split crRNA with different concentrations. At the same time, we also incubated the sample including Cas12a, Cy5-full-sized crRNA, and full-T for 20 min and then added FAM-split crRNA.

As shown in **Fig. 3c**, it could be confirmed that Cy5-full-sized crRNA continues to bind to Cas12a regardless of FAM-split crRNA in the absence of full-T. On contrary, in the presence of full-T, the binding of full-sized crRNA is reduced, and FAM-split crRNA can replace full-sized crRNA to bind to Cas12a. This indicates that the split crRNA, which is initially inhibited by the full-sized crRNA, can reset Cas12a by replacing the full-sized crRNA after the target recognition and cleavage reaction of full-sized crRNA.

As with the reviewer’s comment, we could see the fluorescence signal of Cy5 and FAM were decreased after 30 min without target sequences. However, it seemed not a random crRNA

degradation but the crRNA 5' end trimming progress by LbCas12a¹. That is, it is thought that the fluorophore (Cy5 and FAM) conjugated at the 5'-end of crRNA is removed by LbCas12a.

It was reported that the Cas12a protein harbor an intrinsic endoribonuclease activity that enables it to cleave pre-crRNA directly upstream of the crRNA pseudoknot, different from other CRISPR-Cas systems that rely on standalone ribonucleases for pre-crRNA processing.² In addition, in another previous study, they showed similar gel results that the modified crRNA is trimmed by LbCas12a on its 5'-end in the absence of the activator (the target).¹

To make it clearer, we have performed additional experiment, and added experimental data and more explanation in the revised manuscript (**Lines 138-171**).

3. In several places throughout this work, the authors “put the cart before the horse” and describe a model for how their methods work before presenting the data that supports the model. For example, in the sentence beginning in line 94 states that full-sized and split crRNAs have different binding affinities while the data that prove this statement are not presented until the paragraph beginning in line 129. This pattern of presenting data out of order makes the rational flow of the work difficult to follow. Data should be presented in such a way as to lead the reader to the same logical conclusion that the authors reach.

Response: Thank you for your suggestions. We have carefully revised the manuscript.

4. In this work, the authors describe a new method for performing RNA diagnostics with asymmetric crRNA activities. However, there are serious problems with the organization of the methods section that would make reproducibility challenging. Some of these problems are listed below:

a. Titles of methods sections are not clear enough to unambiguously know to which data within the paper they refer.

b. Methods are not sufficiently detailed to be replicated. Key details like incubation times, concentrations etc. are often omitted.

c. The specifics of how the asymmetric CRISPR assay is designed are not provided.

d. The data presented in figure 2b represent end-point fluorescence measurements, while none of the author provided methods correspond to this experimental set-up. The authors must include the methods of all experiments performed in sufficient detail that they could be replicated.

Response: Thank you for your suggestions. We have added more details in the Method section according to the reviewer's comments a-d in the revised manuscript (**Lines 381-415**).

e. The experiment described in lines 189 – 200, there is no clear method describing how these exact experiments were conducted. Specifically, the source of miR-19a is not clear. Were these miR-19a samples in vitro transcribed or ordered or from patient samples?

Response: The target miRNA (miR-19a) described in previous lines 189 – 200 was synthetic RNA. In the Clinical validation of asymmetric CRISPR assay, miRNAs were extracted from plasma samples of bladder cancer patients. To make it clearer, we have added more details in the revised manuscript (**Lines 292-293; 366-367**).

5. The authors should use the same language to describe their experimental conditions in both the text and figures. For example, in the text description of the competitive crRNA experiment, the

authors use language different from the text in the figure labels. Lack of continuity between the text and figures makes it hard to understand what you are talking about.

Response: Thank you for your suggestion. We have carefully revised the manuscript based on the reviewer's comments.

Minor concerns.

1. The term “asymmetric CRISPR assay” is used liberally throughout, but this terminology is never clearly explained.

Response: We have added more explanation on it in the revised manuscript (**Lines 70-72**).

2. The authors should cite the LEOPARD CRISPR-based detection method in the introduction.

Response: Thank you for the literature. We have cited the reference¹⁸ in the introduction.

3. In line 43, change “developed” to evolved. Developed implies rational design in the emergence of CRISPR-Cas immune systems.

Response: Thanks! Done.

4. Consider removing or clarifying the sentence beginning “Several” on line 55. How much do the kinds of chemical modifications used in the referenced paper really relate to the work at hand.

Response: Thanks! Done.

Since we used split crRNA structurally different from the normal crRNA in this study, we intended to introduce the types of crRNA modification research currently being conducted in relation to CRISPR-based analysis.

5. Consider rewording the sentence beginning on line 59 to read “Another study reported that a split crRNA containing separated scaffold and spacer RNA can catalyze highly specific cis-cleavage in Cas12a, but did not explore the effect on trans-cleavage.”. The following line could then be removed.

Response: Thanks. Done.

6. Clarify dsDNA in line 71, as Cas12a does not require a PAM sequence to be activated by ssDNA.

Response: Thanks! Done.

7. One flaw in existing CRISPR diagnostic methods mentioned in the introduction is the requirement of multiple crRNAs (line 52). Could the authors explain why multiple crRNAs are now beneficial in the authors proposed solution beginning on line 74 (specifically mentioned on line 78).

Response: In existing CRISPR assay without multiple crRNAs, multiple individual crRNAs need to be designed to bind multiple regions with PAM sequences within a target. Thus, it requires that the target sequence itself should be long enough to accommodate binding to multiple crRNAs. On the contrary, in this study, we have employed the split crRNAs and a universal DNA activator,

which are independent of the target sequence, thereby enabling the detection of both long and short targets. We have added some comments in the revised manuscript (**Lines 51-52**).

8. RE. "... PAM-free detection" (line 83). To make this statement, the authors should include data and or rational that supports the PAM independence of the method. Since this is not addressed in the body of the work, it should be omitted.

Response: As suggested by reviewer, we have deleted the statement of PAM-free detection in the revised manuscript.

9. In line 91, remove "(DNA or RNA)" as the illustrated set-up in figure 1 does not show how the authors were able to detect RNA. Further modifications were necessary for RNA detection as illustrated in figure 4.

Response: Thanks! Done.

10. In line 132, the authors describe crRNAs used at different ratios, but the exact concentrations used are not mentioned in the text or methods sections.

Response: The exact concentrations used in this study were mentioned in the **Figure 2** legend. To make it clearer, we have also added the details in the Method section of the revised manuscript.

11. Could the authors explain the use of the phrase cascade signal amplification on lines 140 and 142. Usually, cascade signal amplification refers to mechanisms like phosphorylation cascades or the synthesis of second messengers as seen in type III CRISPR diagnostics (eg. SCOPE). A mechanism of reactivation proposed by the authors does not neatly fit this paradigm and so needs clarification.

Response: We attempted to describe a sequential Cas12a response that proceeds from a full-sized crRNA/CRISPR reaction to a split crRNA/CRISPR reaction, using the term 'cascade reaction', which is commonly used to describe consecutive reactions.

12. Lines 166 – 169. Do the authors perhaps mean lower instead of higher when describing their LOD of 100 fM? The authors should also report a.) the LOD determined for Cas12a w/o split crRNA, and b.) the equation used to interpolate the LOD. Additionally, a horizontal line at the mean blank + 3.3* σ should be included in SF2 for clarity.

Response: Thank you for your suggestion. Yes, we have re-written the sentences and added more details in the revised manuscript (**Lines 184-186**).

As suggested by reviewer, we have added horizontal line at the mean blank + 3.3* σ in the **Supplementary Figure 3** (previous **Supplementary Figure 2**).

13. Line 178 – 181. The sentence beginning on line 178 should be changed to make the experimental rational clearer. For example, "probe" is often used to describe the fluorescent reporter used to measure trans-cleavage, though this is apparently not what the authors mean based on figure 4. Furthermore, the phrase "various locations" is extremely ambiguous. Please change the text to make the meaning clearer.

Response: Thank you for your suggestion, we have changed the term “probe” to “activator”. Also, we have revised the manuscript to make it clearer.

14. Line 180 – 181. Please change the text to clarify what is meant by “when only one type of target is present”.

Response: Thanks! Done.

15. Change “to target” in line 192 to “complimentary to”, for clarity.

Response: Thanks! Done.

16. Can the authors explain why they chose to target different segments of the miR-19a with the RD crRNA vs. DD crRNA? Why not target the same sequence but at the 5’ and 3’ ends? Can the authors also explain why this particular segment of miR-19a was chosen?

Response: Since we selected miR-19a as the target, we wanted to find a way to detect the target miRNA itself. Therefore, we chose to retain the RNA target sequence (miR-19a) and DNA probe (DNA activator) sequence and designed two different DD and RD crRNA. We chose miR-19a as a target because miR-19a is one of the well-known cancer biomarkers. We have added some explanations in the revised manuscript (**Lines 288-289**).

17. The design for detecting RNA presented in figure 5a deviates from the asymmetric CRISPR assay first introduced in figure 1, in that the “DNA-probe” is both complimentary to segments of the full-sized crRNA and the split crRNA. The authors should explain their design rational in the text. Could the authors additionally explain why an independent target for the split crRNA is not used as in figure 1?

Response: The asymmetric CRISPR assay is a signal amplification strategy that uses: i) two types of crRNA (full-sized crRNA and split crRNA), and ii) one DNA target of split crRNA. In other words, **Figure 5a**, which uses two types of crRNA and DNA target of split crRNA also falls into the category of asymmetric CRISPR assay.

After we found that RNA detection is feasible using Cas12a/crRNA in the presence of a DNA probe (currently re-termed as DNA activator), we attempted to apply the asymmetric CRISPR assay to the RNA detection to improve the RNA detection sensitivity. To this end, the DNA activator used for RNA detection was designed to be the target of the split crRNA at the same time, which eliminates the need for additional DNA activator. We have added more discussions in the revised manuscript (**Lines 241-242**).

18. Line 238. Could the authors compare their amplification free results to published data on amplification free diagnostic potential of Cas12a (eg. Ramachandran and Hyuke 2023).

Response: Thank you for the reference. Yes, we have cited this reference ¹⁹ and added some discussion in the revised manuscript (**Lines 235-237**).

19. Line 257 – 258. Did the authors include any blinding controls while testing the bladder cancer patient samples against healthy patients, or did the authors know while performing the assay which samples belonged to which group?

Response: As stated in our Reporting Summary form, no blind test was not conducted in our experiment because the main purpose of this study is to evaluate and compare the performance of the asymmetric CRISPR assay with conventional RT-qPCR.

20. *Are the progress curves presented in Figure 1b experimentally derived? If so, the authors should make that clearer. If not, then these hypothesized outputs should be verified by experimentation.*

Response: It was not experimental data, but a schematic illustration. To avoid any confusion, we have deleted the progress curves and revised the **Figure 1**.

21. *In figure 2c, the authors should present images from the same gel next to each other for easier comparison. Additionally, because two different fluorescent channels are used, the two images could be overlaid for even easier comparison as is common practice when using different fluorophores on nucleic acids (ref).*

Response: Thank you for your suggestions. We have revised the **Figure 2c** by merging gel images measured using different fluorescent channels into a single image.

22. *In figure 3b, the authors should identify all the bands present in the gel. This would be made easier if the authors chose to use denaturing Urea-PAGE or FDF-PAGE gels instead of native page. Could the authors explain why they chose not to image for the FAM label on the split crRNA?*

Response: As suggested by reviewer, we have modified **Figure 3** to show bands more clearly by merging gel images measured with different fluorescence filters. Also, additional FAM labeled-split crRNA gel results were included in **Supplementary Figure 2** as a control group without full-sized crRNA.

Response to Reviewer 3's Comments:

We appreciate reviewer 3 for taking his/her valuable time to comment.

Reviewer #3 (Remarks to the Author):

"I co-reviewed this manuscript with one of the reviewers who provided the listed reports. This is part of the Nature Communications initiative to facilitate training in peer review and to provide appropriate recognition for Early Career Researchers who co-review manuscripts."

Response to Reviewer 4's Comments:

We are grateful to the reviewer 4 for his/her time, enthusiasm, and comments. Below, we reproduce reviewer 4's comment in italics and our response in blue regular print.

Reviewer #4 (Remarks to the Author):

Moon et al. reports an asymmetric CRISPR assay for cascade signal amplification detection of nucleic acids by leveraging the asymmetric trans-cleavage behavior of competitive crRNA, with a narrative that positions the work ultimately for de-centralized diagnostics. As a proof-of-concept, the authors used the CRISPR assay to quantitatively detect a microRNA (miRNA) biomarker for liquid biopsy, demonstrating an amplification-free and PAM-free detection approach using CRISPR-Cas12a.

This is an interesting addition to the CRISPR-based sensor toolbox. The fundamental biology involving the full and split crRNA, and how this can be exploited to develop variable affinity to Cas12a is very interesting and may have broader implications. The authors' use of this phenomenon to create a cascade for signal amplification is creative and the reported LODs are impressive. I'm not able to comment on the novelty of these features in the broader CRISPR literature, but for sensing this represents quite a step toward the practical deployment of CRISPR for point-of-need diagnostics.

While I quite like the manuscript, this is a complex project and there are places in the manuscript where additional text and figures may be required to ensure the material is accessible to wide audience. In summary, the manuscript is well-written, and the contents are also organized and, for the most part, clearly demonstrated. I do have some questions and comments for the authors that are summarized below.

General comments

1. During the review process, I suggest that the authors insert the number of lines in the text to make it easier to reference material.

Response: Thanks! Done.

2. Please check minor spelling errors in the main text (e.g. quantitative, etc).

Response: Thank you for your suggestion! We have carefully revised the manuscript.

3. Statistical analysis (or a description of the analysis in the figure legend) is missing from the data presented (e.g., Fig. 2, 4, 5 and 6). Notably, we can infer some differences based on the values obtained, but I believe that performing adequate statistical analysis between the conditions tested is required for publication. Further, the graphs in figure three do not have error bars (e.g. 4b, 4c, and 4d). Were these experiments replicated?

Similarly, many graphs in the supplemental information (Fig S3-S9) will need the appropriate statistics details added (# replicates, # experiment, SD/error bars).

Response: Following the reviewer's comment, we have added more details on statistical analysis information and replicated experiment in the revised figures and their legends.

4. Figure 1. Authors state "In the presence of target nucleic acids, the Cas12a/full-sized crRNA

complex is specifically activated and initiates the first trans-cleavage reaction, resulting in the cleavage and consumption of full-sized crRNA through Cas12a/crRNA kinetics. Subsequently, the split crRNA can replace the cleaved full-sized crRNA and reactivate Cas12a for the second trans-cleavage reaction, which leads to an additional fluorescence signal amplification”

While it seems intuitive that this would be the case. Have authors tested the addition of a high concentration (e.g. 2x) of full-sized crRNA to confirm that the signal from the asymmetric assay is in fact greater? The abundance of the split-T also would seem to factor into better performance of the asymmetric assay and should probably be mentioned.

Response: Thank you for your suggestion.

As suggested by reviewer, we have tested with higher concentrations of full-sized crRNA. An excessive amount of full-sized crRNA can reduce or inhibit the detection signal (**Figure 5b**). Also, we have optimized the concentration of split-T (DNA activator) (**Figure 5d**). We have added these optimization results and some discussion in the revised manuscript (**Lines 245-258**).

5. Figure 2c. I'm not sure there is a better way to present the data, but it did take a while to properly interpret this figure.

Response: To make it clearer, we have modified **Figure 2c** by merging gel images measured by two different fluorescence filters, Cy5 and Alexa488 into a single image.

6. Figure S1. Please consider labeling the bands on the gel with arrow labels along one side.

Response: Thanks! Done.

7. Figure 3b/c. It would be helpful to label the split and full crRNAs on the side of the gel. It would also be useful to add an arrow to indicate the degradation of the split FAM crRNA.

Response: Thanks! Done.

8. Figure 4. While the concept is very interesting and practically useful, the nomenclature of the fragmented target with probe and target segments was at first quite confusing. This is the first mention of the word “probe” and the concept in general and so deserves some more description. I would also recommend augmenting the schematic to improve understanding (for example, enlarge the images of the probe and/or target with crRNA and Cas12a)

Further, the thin lines and small text in all primary figure schematics makes it difficult to interpret. Please consider increasing line thickness and/or it would be helpful if you can a way to emphasize the critical details that the readers should be tracking.

The seed region (within 5-10 nt PAM) is referenced as an important potential factor. To make the point more accessible to readers, please consider indicating the PAM sequence (e.g. bold or underline).

Response: Thank you for your suggestions. Following the reviewer’s suggestion, we have changed the term “probe” to “activator”. Also, we have carefully revised figures and marked the ‘seed’ region (**Figure 4a**).

9. Figure S3. With the emphasis of the seed region in this data set. Again, marking the seed region and/or the PAM sequence would be helpful.

Response: Thanks! Done.

10. Figure S4. Is the ssDNA probe a critical partner for detection of ssRNA? If so, okay. But the concept needs to be clearly described. Also, what are the consequences to practical implementation? Does the probe have tolerance to SNPs if there is mismatch with the biological target?

Response: Yes, the ssDNA probe (currently re-termed as ssDNA activator) is necessary for RNA detection using Cas12a (**Figure 4**). We have added more details on its concept and implementation in the revised manuscript, including **Figure 4** and **Supplementary Figure 4**.

In our study, we used commercially available EnGen Lba Cas12a from New England Biolabs (NEB, USA), which has been reported to lack single-based resolution for single nucleotide polymorphism (SNP) by previous literature.^{20, 21} We have added some discussion in the revised manuscript (**Lines 353-355**).

Page 10. Authors report “For comparison, we designed a miR-19a 3’-target/ssDNA 5’-probe by reversing the targeted locations of the RD crRNA and designed crRNA that binds to ssDNA 5’-probe and miR-19a 3’-target (DD crRNA). Interestingly, they could not initiate the collateral cleavage reaction of Cas12a regardless of the crRNA binding position (Supplementary Figure 4). By contrast, when both target nucleic acids were ssDNA, we observed an efficient cleavage reaction regardless of the binding length (Supplementary Figure 5). Based on this finding, we placed the miRNA at the 5’-target position close to the 3’ end of the crRNA.”

While data clearly shows that collateral cleavage works if both probe and target are ssDNA (Fig S5) and does not work (for the most part) if the both probe and target are ssDNA and miRNA, respectively (Fig S4). It is not clear how these results guided the authors to place the miRNA at the 5’- target position close to the 3’ end of the crRNA? Some additional description of the rationale would be useful.

Response: We have conducted the experiment using a series of crRNA to bind ssDNA 3’-activator/miR 19a 5’-target with a single-nucleotide difference (previous **Supplementary Figure 3**, current **Supplementary Figure 4**) before we performed the probe (currently re-termed as DNA activator)/ target experiments (previous **Supplementary Figure 4**, current **Supplementary Figure 5**) and ssDNA target detection experiment (previous **Supplementary Figure 5**, current **Supplementary Figure 6**). From these results, we found that the collateral cleavage activity of Cas12a can be activated when miRNA is located at the 3’ end of crRNA. In addition, we could confirm that strong *trans*-cleavage can be initiated when the binding length between crRNA and ssDNA 3’-activator is 12 nt and when the binding length between crRNA and miR 19a 5’-target is 9nt.

11. Figure S6. The authors state “The real-time fluorescence signal was measured depending on various concentrations of target miRNA (100 fM ~ 1 nM) and confirmed that a 1 pM level of miRNA could be detected (Supplementary Figure 6).” In looking at the data, it looks more like the lowest detectible miRNA concentration was 100 pM. How are the authors calculating the limit of detection? It is typically calculated as detection above 3x the SD of the negative control. There are no error bars included and so interpretation of the data is difficult.

Response: The experiment was conducted triplicated, and the LOD was calculated by the equation, mean blank + 3.3 x S.D of a blank (negative control). We have added standard deviation

values and modified the figure in the manuscript as shown in **Supplementary Figure 7** (previous **Supplementary Figure 6**).

12. *Figure S7. The trans-cleavage fluorescence signal presented is very low (~100 RFU) compared to signal in Figure S6 (e.g. 1×10^4 , 1 nM target) and very close to background. Statistical analysis will be important to determine whether the two conditions are similar or just represent background. Author's should include data from high target concentrations to better elucidate the similarity of detection thresholds (ssDNA vs ssDNA/ssRNA).*

Response: Following the reviewer's comment, we have revised the figure to include high-target concentration results and statistical analysis information in **Supplementary Figure 8** (previous **Supplementary Figure 7**).

13. *Figure 5. The description of panel 5a is a helpful summary of how the asymmetric CRISPR assay is rationally designed (pg 10/11). A more detailed description of the design process may help readers onboard the concept into their labs.*

Response: As suggested by reviewer, we have added more explanation in the revised manuscript (**Lines 241-242**).

14. *Figure 5b. Authors state on page 11 "After optimizing the reaction conditions of the split crRNA, DNA probe, and Cas12a in the asymmetric CRISPR assay". It would be important for the readers to understand if these components need to be optimized for each new target sequence or if these represent universal concentrations for the components (with a summary of the conditions selected as optimal).*

Response: To demonstrate its universality, we have further other miRNA (e.g., let-7a) (**Supplementary Figure 10**). In this assay, we utilized the same DNA probe used for miR-19a detection and designed a full-sized crRNA that binds both the let-7a miRNA target and the DNA probe in the same way we designed the crRNA for miR-19a detection. Also, since the same DNA probe was used, the identical split crRNA could be used. We have added some discussion and additional data in the revised manuscript (**Lines 274-276**).

15. *Figure 5 (and S2). The text refers to concentration with femtomolar to nanomolar notation, but the figures use molar with 10^{-x} . It would be good to synchronize these, with my preference to the use of fM, nM, etc as it is more intuitive for readers.*

Response: Thanks! Done.

16. *Figure 5 ef. The authors state "Interestingly, in the high-concentration target miRNA condition, there was no significant difference compared with the CRISPR detection without split crRNA; however, a notable difference in fluorescence signal emerged in the low-concentration target condition." While I agree that the signal difference is less at higher concentrations, there is a strong statistical difference between the two modes of detection at higher concentrations in Figure 5f. Also, the author's present the data as delta FL intensity. Please define the calculation, for instance the change in fluorescence over what period of time?*

Authors state "As shown in Figure 5f, the overall fluorescence signal was enhanced by the addition of split crRNA, and the detection sensitivity was significantly improved compared with

that without split crRNA.” This should read, the “change in fluorescence signal” to reflect the data presented.

Further, it would be important to include the raw data (RFUs) of both modes of detection in the supplemental data.

Response: As suggested by reviewer, we have further defined how we calculated the delta FL intensity in **Figure 5 f** and **g** (previous **Figure 5e** and **f**) in the revised manuscript (**Lines 266-269**). In addition, we have also included the raw RFU data in **Supplementary Figure 9**.

17. *Figure S8. As the author’s highlight, the asymmetric CRISPR assay appears to provide a dose dependent response. However, statistical analysis (I assume SD is indicated by line thickness, but this is not indicated), it is difficult to determine with what level of confidence detection can be assigned. Further, with the RT-qPCR (S9) as contrast, the importance of dynamic range of detection is also key. The difference between control (nc) and the lower concentration target containing samples is quite small.*

It is worth noting that while the reported assay is isothermal and does not require pre-amplification, the current mode of signal generation would require a plate reader to discriminate between positive and negative results in the clinically relevant range of target concentration (femtomolar and below). This is no problem in a proof-of-concept report, but this caveat should be mentioned in the discussion as a challenge that would need to be addressed for deployment as a practical point-of-need diagnostic.

Response: As suggested by reviewer, we have added a statistical analysis to the results in **Supplementary Figure 9** (previous **Supplementary Figure 8**). In addition, we have further discussed potential challenges for point of care diagnostic application in the revised manuscript (**Lines 349-359**).

18. *The authors claim their methodology to be “simple/rapid”, but this is not the case here in my opinion. In fact, making the CRISPR-based assays requires substantial skills, laboratory infrastructure, expensive equipment (e.g. plate reader, real-time PCR instrument), electricity, etc. Therefore, the terms should be used very carefully.*

Response: Following the reviewer’s comment, we have carefully revised the manuscript and discussed the potential challenges (**Lines 349-359**).

19. *Analytical sensitivity experiment.*

Authors state “In particular, the fluorescence intensity linearly increased with the logarithm concentration of miR-19a ranging from 1 fM to 10 pM in the presence of split crRNA. Using the equation $y = 645.4x + 10279.5$ ($R^2 = 0.984$), we estimated the LOD to be 196 aM, which is 5,000 times lower than the CRISPR detection in the absence of split crRNA, which has a LOD of 1 pM”

The authors describe that the developed CRISPR assay was able to detect concentrations around aM. Also, the authors claim that asymmetric CRISPR has a sensitivity of 5,000 times lower than the CRISPR detection in the absence of split crRNA. However, this result is not clear (Fig. S6 and Fig. S8). These low concentrations (e.g. 1 fM, 10 pM) show a low signal close to the control.

The theoretical LOD can not be inferred by an equation, but rather needs to be demonstrated by testing the related target concentrations empirically.

How many times was this experiment repeated? In general, it is common to observe failure in the detection of very low concentrations of nucleic acid.

Have the authors had experience with false positive results?

I suggest authors use probit analysis to determine LOD. This analysis may help answer some of my concerns about LOD.

Response: We used a linear regression curve to calculate the LOD described in previous studies.^{3, 4, 5} In addition, the ICH (International Conference on Harmonization) guideline Q2(R1) on validation of analytical procedures also provides a LOD calculation equation, $LOD = 3.3 \times \sigma / m$ (σ is the standard deviation of the response and m is the slope of the calibration curve).⁶ As suggested by reviewer, we have claimed the LOD based on the lowest concentration we could empirically detected.

In each experiment, the control was always tested together, and the delta fluorescence intensity ($\Delta FL = F_{\text{target}} - F_{\text{control}}$) obtained by subtracting the fluorescence signal of the control from the fluorescence signal of the target is used for analysis. We have added more details in the revised manuscript (**Lines 266-269**).

Initially, the experiment was conducted triplicated. To further test the repeatability and reliability, we have performed additional four tests, and obtained similar results (**Supplementary Figure 9**).

20. Figure 6. It is not clear how the authors correlated the asymmetric CRISPR and RT-qPCR results in figure 6c,d. How did the authors normalize the data from both techniques? This information must be clear in the manuscript.

Response: In the case of RT-qPCR, we first constructed a standard curve (**Supplementary Figure 11**) based on the relationship between miRNA concentration and the Cq values. The Cq values were obtained with the associated software (CFX96 touch real-time PCR system). Then, the obtained Cq values from real samples (patient and healthy donor samples) were substituted into the standard curve to estimate the miRNA concentration.

In the case of asymmetric CRISPR assay, a standard curve (**Figure 5g**) was obtained based on the relationship between miRNA concentration and the Δ Fluorescence intensity ($F_{\text{Target miRNA}} - F_{\text{control}}$). Likewise, the miRNA concentration could be estimated based on the standard curve.

Each miRNA concentration calculated in both ways was compared in **Figure 6**.

21. Figure 6 - I have a concern about two patient samples used in this study. Samples #5 and #8 showed very low signal intensity. Could the authors do statistical analysis comparing these samples with the healthy (negative) samples?

Response: In **Figure 6**, we tried to show that the asymmetric CRISPR assay has a comparable performance with conventional RT-qPCR in clinical sample testing. In patient samples #5 and #8, miRNA concentrations were measured low, however, RT-qPCR results also showed similar results, thus it is not thought to be a problem with miRNA detection performance.

In addition, as suggested by reviewer, we further compared patient samples #5 and #8 with healthy samples #1~#5 by statistical analysis (T-test). We confirmed that both #5 and #8 showed the result of $**P < 0.01$, respectively, indicating statistically significant differences from healthy samples.

22. Additional discussion is needed - The authors provided a concise and brief discussion of the

work, which was well-written. However, with the author's emphasis on de-centralized diagnostic in mind, what would be the cost to perform one reaction using the reported methodology? In terms of application and distribution to low-income countries and many point-of-need applications, this is a critical factor. I believe that the readers would benefit from a calculation for the cost of one reaction, for example. It may also be interesting to address potential future directions in the manuscript.

Response: As suggested by reviewer, we have added a table (**Supplementary Table 2**) showing the cost per reaction. Also, we have added some discussion of potential challenges and applications in resource-poor settings in the revised manuscript (**Lines 349-359**).

23. Additional discussion is needed - I would like to see more content regarding the advantages and barriers to producing and distributing CRISPR-based technologies in low-resource settings (e.g., remote areas, rural cities, etc). What would be the next steps to make the methodology used in the field? What are the bottlenecks to promote decentralization? This was exemplified by the fact that there is no CRISPR-based system being used in reference laboratories for the diagnostics of infectious diseases.

Response: Thank you for your suggestions. We have further discussed its advantages and challenges for use in resource-limited environments^{22, 23} in the revised manuscript (**Lines 344-348; 349-359**).

24. What are the limitations of this study? Possible drawbacks of this approach are not clearly spelled out and should be mentioned in the Discussion.

Response: As suggested by reviewer, we have further discussed the limitations of this study in the Discussion (**Lines 349-359**).

25. Materials and Methods - I have a concern about the methods section. Overall, the authors provided important details for the execution of the methodology. I understand that the methods should be condensed to make it easier to read, but I am favor of transparent science. In my opinion, I suggest the authors provide more details of the methodology in a way that allows the reproduction of the results. These details can be inserted in the supplementary material. Please see supplementary material from the manuscript "Sensitive fluorescence detection of SARS-CoV-2 RNA in clinical samples via one-pot isothermal ligation and transcription" as an example: <https://www.nature.com/articles/s41551-020-00617-5>

Response: Following the reviewer's comment, we have carefully revised the manuscript to add more detailed descriptions in Materials and Methods (**Lines 381-415; 425-432; 444-451**).

26. Patient sample collection - The authors used patient samples in the study and did not provide information how the samples were collected. Please insert these details.

Response: The patient samples were collected and provided by the Carole and Ray Neag Comprehensive Cancer Center with a protocol approved by the ethics committee at the University of Connecticut Health Center (IRB #08-310-1). We have added more detail in the manuscript (**Lines 374-377; 573-574**).

27. Discussion - The authors bring some sentences to the discussion that seem to overvalue the reported data in the manuscript. For example, authors highlight that the reported methodology

“could have important clinical applications in early cancer diagnostics and infectious disease detection”. I suggest authors to be careful with this type of claim, since only one target was used in the current study, nor were any patient trial performed using a large number of clinical samples.

Response: Thank you for your suggestion. We have carefully revised the manuscript (**Lines 359-362**).

Comments in reference to specific lines in the text or figures

1. *Figure 2c and related text. The concentration of the FAM-split crRNA should be inserted in the main figure.*

Response: Thanks! Done.

2. *Figure 3. Please provide the ladder information for each gel.*

Response: Thanks! Done.

3. *RNA extraction and RT-qPCR. Please provide details about the RT-qPCR assay (e.g. volume of reaction, primer concentration, instrument used, oligonucleotide sequences, etc). This will allow trainees to replicate your results. Did the authors use any housekeeping gene in the analysis of the relative miRNA expression?*

Response: Following the reviewer’s comment, we have added more details about RT-qPCR assay in the revised manuscript (**Lines 467-470**).

In our work, we used RT-qPCR standard curves of ten-fold serial dilutions of miR-19a (**supplementary figure 11**) to quantify miRNA because the eligibility of housekeeping genes for miRNA RT-qPCR analysis is limited. In addition, it has been reported that the expression rate of those housekeepers differs among tissue types and inter-individual variations.^{24, 25}

4. *Methods - Please provide details about all kits used in the study (e.g., ID catalog).*

Response: Thanks! Done.

5. *Figure 4b. The legend makes it difficult to see the amplification curve. Please edit.*

Response: Thanks! Done.

6. *Supplementary Figure 2b. Please use the concentration of ssDNA (10 fM ~ 100 pM) instead e.g. -9,-10,-11, etc. This minor change will put the graph more easy for the reader.*

Response: Thanks! Done.

Response to Reviewer 5's Comments:

We appreciate reviewer 5 for taking his/her valuable time to comment.

Reviewer #5 (Remarks to the Author):

"I co-reviewed this manuscript with one of the reviewers who provided the listed reports. This is part of the Nature Communications initiative to facilitate training in peer review and to provide appropriate recognition for Early Career Researchers who co-review manuscripts."

References

1. Nguyen LT, Smith BM, Jain PK. Enhancement of trans-cleavage activity of Cas12a with engineered crRNA enables amplified nucleic acid detection. *Nat Commun* **11**, 4906 (2020).
2. Swarts DC, van der Oost J, Jinek M. Structural Basis for Guide RNA Processing and Seed-Dependent DNA Targeting by CRISPR-Cas12a. *Mol Cell* **66**, 221-233 e224 (2017).
3. Hajian R, *et al.* Detection of unamplified target genes via CRISPR-Cas9 immobilized on a graphene field-effect transistor. *Nat Biomed Eng* **3**, 427-437 (2019).
4. Song J, *et al.* Elution-free DNA detection using CRISPR/Cas9-mediated light-up aptamer transcription: Toward all-in-one DNA purification and detection tube. *Biosens Bioelectron* **225**, 115085 (2023).
5. Suea-Ngam A, Howes PD, deMello AJ. An amplification-free ultra-sensitive electrochemical CRISPR/Cas biosensor for drug-resistant bacteria detection. *Chem Sci* **12**, 12733-12743 (2021).
6. Borman P, Elder D. Q2(R1) Validation of Analytical Procedures. In: *ICH Quality Guidelines* (2017).
7. Rananaware SR, *et al.* Programmable RNA detection with CRISPR-Cas12a. *bioRxiv*, (2023).
8. Hellman LM, Fried MG. Electrophoretic mobility shift assay (EMSA) for detecting protein-nucleic acid interactions. *Nat Protoc* **2**, 1849-1861 (2007).
9. Wang F, Yao T, Yang W, Wu P, Liu Y, Yang B. Protocol to detect nucleotide-protein interaction in vitro using a non-radioactive competitive electrophoretic mobility shift assay. *STAR Protoc* **3**, 101730 (2022).
10. Bezzetti V, Borgatti M, Finotti A, Tamanini A, Gambari R, Cabrini G. Mapping the transcriptional machinery of the IL-8 gene in human bronchial epithelial cells. *J Immunol* **187**, 6069-6081 (2011).
11. Zhong M, *et al.* PCDetection: PolyA-CRISPR/Cas12a-based miRNA detection without PAM restriction. *Biosens Bioelectron* **214**, 114497 (2022).
12. Jia HY, Zhao HL, Wang T, Chen PR, Yin BC, Ye BC. A programmable and sensitive CRISPR/Cas12a-based MicroRNA detection platform combined with hybridization chain reaction. *Biosens Bioelectron* **211**, 114382 (2022).
13. Yan H, *et al.* A one-pot isothermal Cas12-based assay for the sensitive detection of microRNAs. *Nat Biomed Eng*, (2023).
14. Peng S, Tan Z, Chen S, Lei C, Nie Z. Integrating CRISPR-Cas12a with a DNA circuit as a generic sensing platform for amplified detection of microRNA. *Chem Sci* **11**, 7362-7368 (2020).
15. Li X, *et al.* Ultrasensitive detection of microRNAs based on click chemistry-terminal deoxynucleotidyl transferase combined with CRISPR/Cas12a. *Biochimie* **208**, 38-45 (2023).
16. Zhang M, Wang H, Wang H, Wang F, Li Z. CRISPR/Cas12a-Assisted Ligation-Initiated Loop-Mediated Isothermal Amplification (CAL-LAMP) for Highly Specific Detection of microRNAs. *Anal Chem* **93**, 7942-7948 (2021).

17. Zhang G, Zhang L, Tong JT, Zhao XX, Ren JL. CRISPR-Cas12a enhanced rolling circle amplification method for ultrasensitive miRNA detection. *Microchem J* **158**, 105239 (2020).
18. Jiao C, *et al.* Noncanonical crRNAs derived from host transcripts enable multiplexable RNA detection by Cas9. *Science* **372**, 941-948 (2021).
19. Huyke DA, Ramachandran A, Bashkirov VI, Kotseroglou EK, Kotseroglou T, Santiago JG. Enzyme Kinetics and Detector Sensitivity Determine Limits of Detection of Amplification-Free CRISPR-Cas12 and CRISPR-Cas13 Diagnostics. *Anal Chem* **94**, 9826-9834 (2022).
20. Zhang HX, *et al.* Cas12a-based one-pot SNP detection with high accuracy. *Cell Insight* **2**, 100080 (2023).
21. Wu Y, *et al.* A PAM-free CRISPR/Cas12a ultra-specific activation mode based on toehold-mediated strand displacement and branch migration. *Nucleic Acids Res* **50**, 11727-11737 (2022).
22. Huang Z, Lyon CJ, Hu TY. CRISPR-based assays for low-resource settings. *Nat Rev Bioeng* **1**, 230-231 (2023).
23. Land KJ, Boeras DI, Chen XS, Ramsay AR, Peeling RW. REASSURED diagnostics to inform disease control strategies, strengthen health systems and improve patient outcomes. *Nat Microbiol* **4**, 46-54 (2019).
24. Link F, Krohn K, Schumann J. Identification of stably expressed housekeeping miRNAs in endothelial cells and macrophages in an inflammatory setting. *Sci Rep* **9**, 12786 (2019).
25. Xiang M, *et al.* U6 is not a suitable endogenous control for the quantification of circulating microRNAs. *Biochem Biophys Res Commun* **454**, 210-214 (2014).

Reviewers' Comments:

Reviewer #1:

Remarks to the Author:

In this resubmission, the authors have improved the manuscript significantly by providing new supporting data and careful revision of the main text. All previous comments have been addressed.

Reviewer #2:

Remarks to the Author:

The authors have satisfactorily addressed my previous concerns and I believe the manuscript is significantly improved and should be accepted for publication after addressing some minor suggestions listed below:

1. To help with clarity, On Figure 2a and 2b it would be helpful to add diagrams that indicate the labeled full-T and split T targets or crRNAs as was done in panel 2c for the labeled crRNA. It would also be helpful to show the labeled full-T or split T as a diagram.

To help with clarity, On Figure 2a and 2b it would be helpful to add diagrams that indicate the labeled full-T and split T targets or crRNAs as was done in panel 2c for the labeled crRNA. It would also be helpful to show the labeled full-T or split T as a schematic.

2. Line 205 should read similar finding instead of similar founding. Please double check manuscript for additional grammatical or spelling mistakes.

3. Ref. 31 is now published: Rananaware, S.R., Vesco, E.K., Shoemaker, G.M. et al. Programmable RNA detection with CRISPR-Cas12a. Nat Commun 14, 5409 (2023). <https://doi-org.dist.lib.usu.edu/10.1038/s41467-023-41006-1>

4. In figure 5 add the label target miRNA next to the 10 pM label.

5. I am in agreement with the student reviewer's concern 1. below

Comment not necessary to address for publication: I am very intrigued by the authors results showing heightened collateral cleavage and diagnostic sensitivity in the presence of split-crRNA and split-T. While I do not think it necessary for this manuscript to get published I am very curious if the authors have tried their assay with only the spacer-split-T duplex, without the split crRNA-guide. If so, I think the results would be of value to add to the manuscript. Such data would help elucidate the full mechanism behind this interesting observation. Additionally, kinetic experiments demonstrating how the split substrates effect rates of collateral cleavage would be very insightful.

Comments from a graduate student reviewer;

The authors have sufficiently responded to many of the concerns raised during the review. The manuscript is greatly improved, especially in clarity. However, there are a few concerns that need to be addressed before the manuscript can be recommended for publication:

Minor Concerns:

1. While the authors have modified the text discussing the limits of detection obtained with their split CRISPR method, "higher" in lines 186 and 272 should be changed to "more sensitive" to accurately describe the data presented in Supplemental Figure 3 and 9, respectively.

2. While the authors did label their bands in Figures 3a and 3b, some of the labels do not make sense. For example, the authors have labeled a band of higher molecular weight than Full-T and Split-T as cleavage products. (It is our hypothesis that these bands are RNA/DNA hybrids formed from the spacer/full crRNA and complementary targets). Furthermore, the authors have not

responded to why they chose to examine cleavage reactions with native gels instead of more common practice of denaturing gels.

Reviewer #3:

Remarks to the Author:

"I co-reviewed this manuscript with one of the reviewers who provided the listed reports. This is part of the Nature Communications initiative to facilitate training in peer review and to provide appropriate recognition for Early Career Researchers who co-review manuscripts."

Reviewer #4:

Remarks to the Author:

I thank the authors for sharing an improved, revised manuscript. I praise the authors for addressing most of my concerns, but some of my initial concerns were not sufficiently addressed. I believe that addressing these remaining concerns will strengthen the conclusions and statements made regarding the assay's performance.

I do have some questions and comments for the authors that are summarized below.

Analytical sensitivity analysis

I still have questions about the LOD analysis. The authors reported in the first version of the manuscript that the system was able to detect aM concentrations. In the present version, the authors reported that the LOD was 1fM. I understand that the authors had initially calculated the LOD based on an equation, but I strongly suggest that the authors do probit analysis. Repeating 1 fM ten times, how many times is the system able to detect this concentration? As I said previously, molecular diagnostic systems are prone to failing to detect low concentrations of the target, even RT-qPCR. Probit analysis will provide more robustness for this analysis. Also, I would like to have seen this LOD change more transparently documented during the review process as it is a critical information for understanding the statements reported in the present manuscript and the potential impact of the technology. This information was edited in the abstract, yet was not highlighted with a track change in the revised manuscript.

I also suggest that figure S9 be moved to the main text as it is one of the main findings reported in the manuscript.

Correlation between the the asymmetric CRISPR and RT-qPCR results (Figure 6c,d).

This information must be clear in the manuscript. I would have liked to have seen the response provided in the response letter inserted in the manuscript. I think this information is useful for readers. Also, please use RT-qPCR instead of RT-PCR (Figure 6c,d).

Quality of figures

I recognize the work done by the authors to improve the quality of the figures, but I suggest the authors provide high resolution figures (e.g. at least 300 dpi) for the next round of revisions. The figures are key elements for understanding the manuscript. Perhaps this is an artifact of uploading the figures.

Reviewer #5:

Remarks to the Author:

"I co-reviewed this manuscript with one of the reviewers who provided the listed reports. This is part of the Nature Communications initiative to facilitate training in peer review and to provide

appropriate recognition for Early Career Researchers who co-review manuscripts.”

Response to Reviewer 1's Comments:

We appreciate reviewer 1 for taking his/her valuable time to comment. Below, we have presented reviewer 1's comment in italics, and the corresponding responses are marked in blue font.

Reviewer #1 (Remarks to the Author):

In this resubmission, the authors have improved the manuscript significantly by providing new supporting data and careful revision of the main text. All previous comments have been addressed.

Response: Thank you for your positive comments!

Response to Reviewer 2's Comments:

We appreciate reviewer 2 for taking his/her valuable time to comment. Below, we have presented reviewer 2's comment in italics, and the corresponding responses are marked in blue font.

Reviewer #2 (Remarks to the Author):

The authors have satisfactorily addressed my previous concerns and I believe the manuscript is significantly improved and should be accepted for publication after addressing some minor suggestions listed below:

Response: Thank you for your positive comments!

1. To help with clarity, On Figure 2a and 2b it would be helpful to add diagrams that indicate the labeled full-T and split T targets or crRNAs as was done in panel 2c for the labeled crRNA. It would also be helpful to show the labeled full-T or split T as a diagram.

To help with clarity, On Figure 2a and 2b it would be helpful to add diagrams that indicate the labeled full-T and split T targets or crRNAs as was done in panel 2c for the labeled crRNA. It would also be helpful to show the labeled full-T or split T as a schematic.

Response: Thank you for your comment. In **Figures 2a** and **2b**, we used unlabeled crRNAs and unlabeled targets, not labeled crRNAs and targets. But we did use the labeled full-T and split-T in **Figure 3a**. Thus, we have added diagrams of the labeled full-T and split-T, and revised **Figure 3**.

2. Line 205 should read similar finding instead of similar founding. Please double check manuscript for additional grammatical or spelling mistakes.

Response: Thank you for your comments. We have carefully corrected it in the revised manuscript.

3. Ref. 31 is now published: Rananaware, S.R., Vesco, E.K., Shoemaker, G.M. et al. Programmable RNA detection with CRISPR-Cas12a. Nat Commun 14, 5409 (2023). <https://doi-org.dist.lib.usu.edu/10.1038/s41467-023-41006-1>

Response: Thank you for your update of the reference 31. We have updated it in the revised manuscript (**Lines 688-689**).

4. In figure 5 add the label target miRNA next to the 10 pM label.

Response: Thank you for your suggestion. We have changed the figure label in **Figure 5** in the revised manuscript.

5. I am in agreement with the student reviewer's concern 1. Below

Response: Thank you for your suggestion. Done!

Comment not necessary to address for publication: I am very intrigued by the authors results showing heightened collateral cleavage and diagnostic sensitivity in the presence of split-crRNA and split-T. While I do not think it necessary for this manuscript to get published I am very curious if the authors have tried their assay with only the spacer-split-T duplex, without the split crRNA-guide. If so, I think the results would be of value to add to the manuscript. Such data would help elucidate the full mechanism behind this interesting observation. Additionally, kinetic experiments demonstrating how the split substrates effect rates of collateral cleavage would be very insightful.

Response: Thank you for your comments. Although the reviewer pointed out that “Comment not necessary to address for publication”, we have still performed additional experiment to compare the CRISPR/Cas12a assay under different conditions of split crRNA (handle and spacer). We could confirm that either split crRNA handle or spacer did not improve the fluorescence signal when compared to the conventional CRISPR/Cas12a assay (**Supplementary Figure 4**). In other words, the improved fluorescence signal was only observed in the asymmetric CRISPR assay containing both split crRNA handle and spacer. Also, we have added some discussion in the revised manuscript (**Lines 184-185**).

Comments from a graduate student reviewer;

The authors have sufficiently responded to many of the concerns raised during the review. The manuscript is greatly improved, especially in clarity. However, there are a few concerns that need to be addressed before the manuscript can be recommended for publication:

Response: Thank you for your positive comments!

Minor Concerns:

1. *While the authors have modified the text discussing the limits of detection obtained with their split CRISPR method, “higher” in lines 186 and 272 should be changed to “more sensitive” to accurately describe the data presented in Supplemental Figure 3 and 9, respectively.*

Response: Thank you for your suggestion. Done!

2. *While the authors did label their bands in Figures 3a and 3b, some of the labels do not make sense. For example, the authors have labeled a band of higher molecular weight than Full-T and Split-T as cleavage products. (It is our hypothesis that these bands are RNA/DNA hybrids formed from the spacer/full crRNA and complementary targets). Furthermore, the authors have not responded to why they chose to examine cleavage reactions with native gels instead of more common practice of denaturing gels.*

Response: In **Figure 3a**, we were able to observe several bands corresponding to Cy5-containing full-T fragments, which varied with the reaction time. The length of the initial Cy5-full-T is relatively long (53 nt) therefore, it is thought to show a cleavage pattern by CRISPR/Cas12a. As the reviewer pointed out, some of the bands were located in higher molecular weight than full-T and split-T. We attribute this aberrant electrophoretic migration to the positive charge of the Cy5 dye. In previous literature¹, it has been reported that 5'-Cy5 labeled oligonucleotides degradation products, ranging from one to eight nucleotides (nt), showed the unusual pattern mobility. In addition, in **Supplementary Figure 1a**, we were able to observe that the cleaved Cy5-full-T fragment was placed at a higher molecular weight

position in the reaction between Cas12a/full-sized crRNA and Cy5-full-T, without split crRNA and split-T.

Here, we consider native gels to be more suitable for identifying hybrid structures between crRNA and target DNAs than denaturing gels, which are more useful for analyzing or separating single-stranded DNA or RNA fragments.

Response to Reviewer 3's Comments:

We appreciate reviewer 3 for taking his/her valuable time to comment.

Reviewer #3 (Remarks to the Author):

"I co-reviewed this manuscript with one of the reviewers who provided the listed reports. This is part of the Nature Communications initiative to facilitate training in peer review and to provide appropriate recognition for Early Career Researchers who co-review manuscripts."

Response to Reviewer 4's Comments:

We appreciate reviewer 4 for taking his/her valuable time to comment. Below, we have presented reviewer 4's comment in italics, and the corresponding responses are marked in blue font.

Reviewer #4 (Remarks to the Author):

I thank the authors for sharing an improved, revised manuscript. I praise the authors for addressing most of my concerns, but some of my initial concerns were not sufficiently addressed. I believe that addressing these remaining concerns will strengthen the conclusions and statements made regarding the assay's performance.

Response: Thank you for your positive comments!

I do have some questions and comments for the authors that are summarized below.

1. Analytical sensitivity analysis

I still have questions about the LOD analysis. The authors reported in the first version of the manuscript that the system was able to detect aM concentrations. In the present version, the authors reported that the LOD was 1fM. I understand that the authors had initially calculated the LOD based on an equation, but I strongly suggest that the authors do probit analysis. Repeating 1 fM ten times, how many times is the system able to detect this concentration? As I said previously, molecular diagnostic systems are prone to failing to detect low concentrations of the target, even RT-qPCR. Probit analysis will provide more robustness for this analysis. Also, I would like to have seen this LOD change more transparently documented during the review process as it is a critical information for understanding the statements reported in the present manuscript and the potential impact of the technology. This information was edited in the abstract, yet was not highlighted with a track change in the revised manuscript.

Response: Thank you for your comments. We have performed additional experiment and conducted a probit analysis in the revised manuscript (**Supplementary Table 1**). Also, we have highlighted the changes in the abstract of the revised manuscript.

2. I also suggest that figure S9 be moved to the main text as it is one of the main findings reported in the manuscript.

Response: Thank you for your comments. We apologize for any confusion. In fact, the main finding/data of previous **Supplementary Figure 9** (current **Supplementary Figure 10**) has been incorporated in **Figure 5g**.

3. Correlation between the asymmetric CRISPR and RT-qPCR results (Figure 6c,d).

This information must be clear in the manuscript. I would have liked to have seen the response provided in the response letter inserted in the manuscript. I think this information is useful for readers. Also, please use RT-qPCR instead of RT-PCR (Figure 6c,d).

Response: Thank you for your comments. We have revised the manuscript to insert the

information (**Lines 297-306**). Also, we have updated **Figures 6c** and **6d** as suggested.

4. Quality of figures

I recognize the work done by the authors to improve the quality of the figures, but I suggest the authors provide high resolution figures (e.g. at least 300 dpi) for the next round of revisions. The figures are key elements for understanding the manuscript. Perhaps this is an artifact of uploading the figures.

Response: Thank you for the suggestion. We have provided high resolution figures in the revised manuscript.

Response to Reviewer 5's Comments:

We appreciate reviewer 5 for taking his/her valuable time to comment.

Reviewer #5 (Remarks to the Author):

"I co-reviewed this manuscript with one of the reviewers who provided the listed reports. This is part of the Nature Communications initiative to facilitate training in peer review and to provide appropriate recognition for Early Career Researchers who co-review manuscripts."

References

1. Killelea T, Saint-Pierre C, Ralec C, Gasparutto D, Henneke G. Anomalous electrophoretic migration of short oligodeoxynucleotides labelled with 5'-terminal Cy5 dyes. *Electrophoresis* **35**, 1938-1946 (2014).

Reviewers' Comments:

Reviewer #4:

Remarks to the Author:

I thank the authors for sharing an improved, revised manuscript. I praise the authors for addressing most of my concerns, but I still have a few minor questions before recommending this manuscript for publication. I believe that addressing these remaining concerns will strengthen the statements made regarding the assay's performance.

I do have some questions and comments for the authors that are summarized below.

Probit analysis

I still have questions about the LOD analysis. The authors have provided detailed information about the LOD (Supplementary Table 1), but the probit analysis has not yet been done. This analysis will answer the lowest concentration that the test is capable of detecting with a confidence interval of 95%. The authors can use MedCalc software to calculate or any other software. I strongly suggest that authors carry out this analysis to check whether the LOD is 1 fM.

Standard deviation (SI figures)

My another comment is that some of the SI figure still look like there is no SD included in the graphs (Supplementary Figure 9, b-c [kinetic graphs]; Supplementary Figure 10, a-b [kinetic graphs]). This information must be included.

Reviewer #5:

Remarks to the Author:

"I co-reviewed this manuscript with one of the reviewers who provided the listed reports. This is part of the Nature Communications initiative to facilitate training in peer review and to provide appropriate recognition for Early Career Researchers who co-review manuscripts."

Response to Reviewer 4's Comments:

We appreciate reviewer 4 for taking his/her valuable time to comment. Below, we have presented reviewer 4's comment in italics, and the corresponding responses are marked in blue font.

Reviewer #4 (Remarks to the Author):

I thank the authors for sharing an improved, revised manuscript. I praise the authors for addressing most of my concerns, but I still have a few minor questions before recommending this manuscript for publication. I believe that addressing these remaining concerns will strengthen the statements made regarding the assay's performance.

I do have some questions and comments for the authors that are summarized below.

1. Probit analysis

I still have questions about the LOD analysis. The authors have provided detailed information about the LOD (Supplementary Table 1), but the probit analysis has not yet been done. This analysis will answer the lowest concentration that the test is capable of detecting with a confidence interval of 95%. The authors can use MedCalc software to calculate or any other software. I strongly suggest that authors carry out this analysis to check whether the LOD is 1 fM.

Response: Thank you for your suggestion. We have done the probit analysis (**Supplementary Table 1**) and further discussed it in the manuscript.

2. Standard deviation (SI figures)

My another comment is that some of the SI figure still look like there is no SD included in the graphs (Supplementary Figure 9, b-c [kinetic graphs]; Supplementary Figure 10, a-b [kinetic graphs]). This information must be included.

Response: Thank you for your suggestion! We have included the information and revised the Figures.

Response to Reviewer 5's Comments:

We appreciate reviewer 5 for taking his/her valuable time to comment.

Reviewer #5 (Remarks to the Author):

"I co-reviewed this manuscript with one of the reviewers who provided the listed reports. This is part of the Nature Communications initiative to facilitate training in peer review and to provide appropriate recognition for Early Career Researchers who co-review manuscripts."